# A Suite of Generative Tasks for
# Multi-Level Multimodal Webpage Understanding

**Andrea Burns**[1][*] **Krishna Srinivasan**[2] **Joshua Ainslie**[2] **Geoff Brown**[2]

**Bryan A. Plummer**[1] **Kate Saenko**[1,3] **Jianmo Ni**[2] **Mandy Guo**[2]

[1]Boston University, [2]Google, [3]FAIR

{aburns4, bplum, saenko}@bu.edu, {krishnaps, jainslie, geoffbrown, jianmon, xyguo}@google.com

## Abstract

Webpages have been a rich, scalable resource for vision-language and language only tasks. Yet only pieces of webpages are kept in existing datasets: image-caption pairs, long text articles, or raw HTML, never all in one place. Webpage tasks have resultingly received little attention and structured image-text data left underused. To study multimodal webpage understanding, we introduce the Wikipedia Webpage suite (WikiWeb2M) containing 2M pages with all of the associated image, text, and structure data[1]. We verify its utility on three generative tasks: page description generation, section summarization, and contextual image captioning. We design a novel attention mechanism Prefix Global, which selects the most relevant image and text content as global tokens to attend to the rest of the webpage for context. By using page structure to separate such tokens, it performs better than full attention with lower computational complexity. Extensive experiments show that the new data in WikiWeb2M improves task performance compared to prior work.

## 1 Introduction

Webpages are a source of multimodal, structured content which have been used for both pretraining and finetuning purposes. Large scale noisy text or multimodal datasets scraped from the web have been used to pretrain large language or contrastive models (Raffel et al., 2020; Jia et al., 2021; Radford et al., 2021; Aghajanyan et al., 2022). Downstream tasks built from webpages have included instruction following, image captioning, news captioning, image-sentence retrieval, and image-article retrieval (Shi et al., 2015; Li et al., 2020; Gur et al., 2022; Sharma et al., 2018; Biten et al., 2019; Liu et al., 2021; Srinivasan et al., 2021; Tan et al., 2022).

---

[*]Work done during an internship at Google.

[1]Data can be downloaded at https://github.com/google-research-datasets/wit/blob/main/wikiweb2m.md.

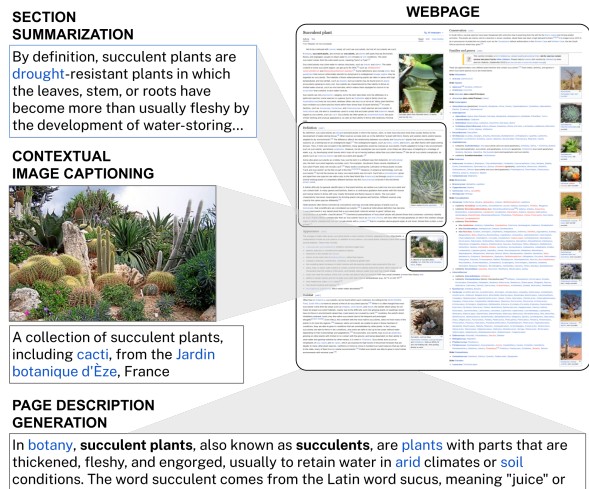

**SECTION SUMMARIZATION**
By definition, succulent plants are drought-resistant plants in which the leaves, stem, or roots have become more than usually fleshy by the development of water-storing...

**CONTEXTUAL IMAGE CAPTIONING**
A collection of succulent plants, including cacti, from the Jardin botanique d'Èze, France

**WEBPAGE**

**PAGE DESCRIPTION GENERATION**
In botany, **succulent plants**, also known as **succulents**, are plants with parts that are thickened, fleshy, and engorged, usually to retain water in arid climates or soil conditions. The word succulent comes from the Latin word sucus, meaning "juice" or "sap". Succulent plants may store water in various structures, such as leaves and stems. The water content of some succulent organs can get up to 90–95%...

Figure 1: Tasks we study with WikiWeb2M. Our dataset provides a unified webpage sample that contains all text, image, and structure, enabling new tasks like page description generation. For image captioning and section summarization, remaining page text and images provide useful context, aiding task performance.

Yet limited prior work has studied tasks to evaluate multimodal webpage understanding itself.

Many classification and generation problems can be studied with webpages: taxonomic webpage classification, webpage retrieval, web image captioning, and page summarization. However, to date there is no open source, multimodal dataset that retains all webpage content. *E.g.*, the Wikipedia Image Text (WIT) dataset (Srinivasan et al., 2021) does not retain HTML structure and misses out on text sections. Thus, we propose the new Wikipedia Webpage (WikiWeb2M) dataset of over 2M pages, which unifies webpage content to include all text, images, and their location (*e.g.*, section index) in a single sample. Table 1 compares the statistics of WikiWeb2M to the existing English WIT dataset.

Figure 1 shows an example of our WikiWeb2M benchmark suite; we design a set of tasks that require webpage understanding at varying degrees of

| Dataset | # Webpage Sections | | | | | | # Images | |
|---|---|---|---|---|---|---|---|---|
| | Structural | Heading | Text | Image | Both | Total | Unique | Total |
| WIT (En) | - | - | - | 199,872 | 2,847,929 | 3,047,801 | 3,660,211 | 4,955,835 |
| WikiWeb2M | 731,394 | 686,376 | 6,817,950 | 221,523 | 3,236,254 | 11,693,497 | 4,438,642 | 5,940,431 |

Table 1: WikiWeb2M versus WIT (Srinivasan et al., 2021). WikiWeb2M re-introduces millions of text and multimodal webpage sections. We report counts over all splits; train, validation, and test are reported separately in Appendix A. WikiWeb2M and WIT (English subset) come from the same webpages.

granularity. Specifically, we use page description generation, section summarization, and contextual image captioning to evaluate a model's ability to understand a webpage at a global, regional, and local level, respectively. For page description generation, the goal is to generate an overarching global description of the webpage. The task of section summarization generates a sentence that captures the key content of one section. Finally, contextual image captioning generates a caption for one image within the webpage.

WikiWeb2M's tasks will allow for general study of multimodal content understanding with many-to-many text and image relationships and can also specifically improve interaction with web content. For example, a webpage description may provide a user who is blind more agency by allowing them to preview content before listening to the entire body of image and text with a screen reader (Vtyurina et al., 2019). In addition to contextual captioning and section summarization aiding assistive technology, these tasks can be used for modern content generation, as there is growing interest in providing multimodal web snippets (Nkemelu et al., 2023). The study of webpages in a multimodal context has even been motivated from a sociological and anthropological perspective (Pauwels, 2012).

While we curate a new dataset with Wikipedia, we note it is just one of many domains that could be used to study multimodal webpage understanding. Instructional websites, news articles, recipes, blogs, and more have bodies of text and images interleaved by layout or HTML structure.

We utilize the T5 (Raffel et al., 2020) framework to address the WikiWeb2M tasks. One challenge in modeling webpage tasks is the length of the input data (*i.e.*, a long sequence results from flattening webpage text and images). While the full attention originally used in T5 is performant, it results in a quadratic computational complexity with respect to the input sequence length. Thus, we define a new mixture of local-global attention, Prefix Global,

which uses our structured webpage data to select the most salient text and images as global tokens in the prefix of our input sequence. Prefix Global is ultimately more efficient, meaning longer input sequences can be used to reach better task performance. Our results can be beneficial to the many structured image-text domains outside of webpages such as mobile apps, figures, posters, infographics, and documents.

We include ablations across multiple axes: the pretrained checkpoint we initialize from, the input sequence length, the feature inputs, and the attention mechanism. We importantly find that images improve performance for all tasks, while prior work on contextual image captioning claimed otherwise (Nguyen et al., 2022). We are also able to improve task performance now that we have access to the entire page's content. Still, there is plenty of room to improve upon our benchmark suite.

We summarize our contributions below:

- A new open source multimodal webpage dataset, WikiWeb2M, containing 2M pages curated from English Wikipedia articles. Each sample contains all text, images, and structure present per page.

- A suite of multimodal generation webpage tasks that reflect webpage understanding at three granularities: page description, section summarization, contextual image captioning.

- A new attention mechanism, Prefix Global, which is a mixture of local-global attention that separates a prefix of global tokens. By defining more salient content from structured pages, it can outperform full attention while requiring fewer attention computations.

- Ablations on attention, sequence length, input features, and model size. Images can help all tasks, notably by over 15% on contextual captioning, and page context boosts average performance by over 4% and 3% for section summarization and captioning, respectively.

| WikiWeb2M | Train | Val | Test |
|---|---|---|---|
| # Pages | 1,803,225 | 100,475 | 100,833 |
| # Sections | 10,519,294 | 585,651 | 588,552 |
| # Total Images | 5,340,708 | 299,057 | 300,666 |

Table 2: Breakdown of the number of pages, sections, and images contained in each WikiWeb2M dataset split.

| Task | Train | Val | Test |
|---|---|---|---|
| Page Desc. | 1,435,263 | 80,103 | 80,339 |
| Section Summ. | 3,082,031 | 172,984 | 173,591 |
| Image Caption. | 2,222,814 | 124,703 | 124,188 |

Table 3: Number of samples for page description generation, section summarization, and image captioning task datasets after additional filtering of WikiWeb2M.

## 2 The WikiWeb2M Dataset

We create the Wikipedia Webpage (WikiWeb2M) dataset to have an all-in-one multimodal webpage dataset where all text, image, and structure content is retained. WikiWeb2M is built by starting with the Wikipedia Image Text (WIT; Srinivasan et al., 2021) English pages[2]. We re-scrape webpage samples and keep all text, image, and structure available, providing more contextual data which can be used to model existing tasks like contextual image captioning, as well as enable new webpage understanding tasks like page description generation. We start with WIT URLs to create a high quality multimodal webpage dataset that has already gone through extensive content and topic filtering.

Each webpage sample includes the page URL, page title, section titles, section text, images and their captions, and indices for each section, their parent section, their children sections, and more. This differs from WIT, which defined individual samples as image-caption pairs with metadata (*e.g.*, originating section title). Appendix A.3 includes a comparison of fields available in WikiWeb2M versus WIT. In Table 1, we report the number of sections and images compared to the English subset of WIT. We add nearly 1M total images to the dataset by keeping the images on a webpage regardless of whether they have image captions available.

We provide section counts by type: structural, heading, text only, image only, and both text and image. Structural and heading sections do not contain immediate text. The former has subsections. For heading sections, a section title was available, while the content linked to a different article, was empty, or only had tables. We only retain sections if they are content sections (*e.g.*, not the "See Also" section). A significant 6.8M text sections are in WikiWeb2M, none of which were available in WIT. For image quality control, we keep JPEG and PNG image types[3]. We make a random 90/5/5 split and

show the number of pages, sections, and images per split in Table 2. Note that Table 2 reflects statistics of WikiWeb2M, which is later refined to build our downstream tasks datasets. It can be repurposed for other webpage understanding tasks or reprocessed with different data filters.

### 2.1 The WikiWeb2M Tasks

We apply WikiWeb2M to three tasks which reflect different granularities of webpage understanding: the page, section, or element level. Table 3 contains task sample counts which are achieved by further task-specific filtering; these data processing steps are included in Appendix A.1, with a discussion of potential dataset noise in Appendix A.2. We describe the tasks below.

**Page Description Generation.** In the task of page description generation, the goal is to generate a description of a page given the rest of the webpage's image, text, and structure. We use the Wikipedia-provided page description (not collecting annotations) and generate summaries from multimodal inputs, which differs from existing text-only article summarization work; this matters when we want to create a multimodal snippet from a webpage.

**Section Summarization.** The goal of section summarization is to generate a single sentence that highlights a particular section's content. The summary is generated given all images and (non-summary) text present in the target and context sections; see Figure 3 for a task example. Following the leading sentence bias, we use the first sentence of a section as a pseudo summary (which is removed from the model inputs). We also found that a majority of human annotators deemed the first sentence as a reasonable summary; these findings are later discussed in Appendix F.

**Contextual Image Captioning.** Nguyen et al. (2022) proposed contextual image captioning with WIT as the task of captioning an image given the image and its webpage context. Target images are those available in WIT to ensure they have quality

---
[2]WIT held a CC BY-SA 3.0 license, and additional data we recover in WikiWeb2M is publicly available on Wikipedia.
[3]We release image URLs, where they can be fetched.

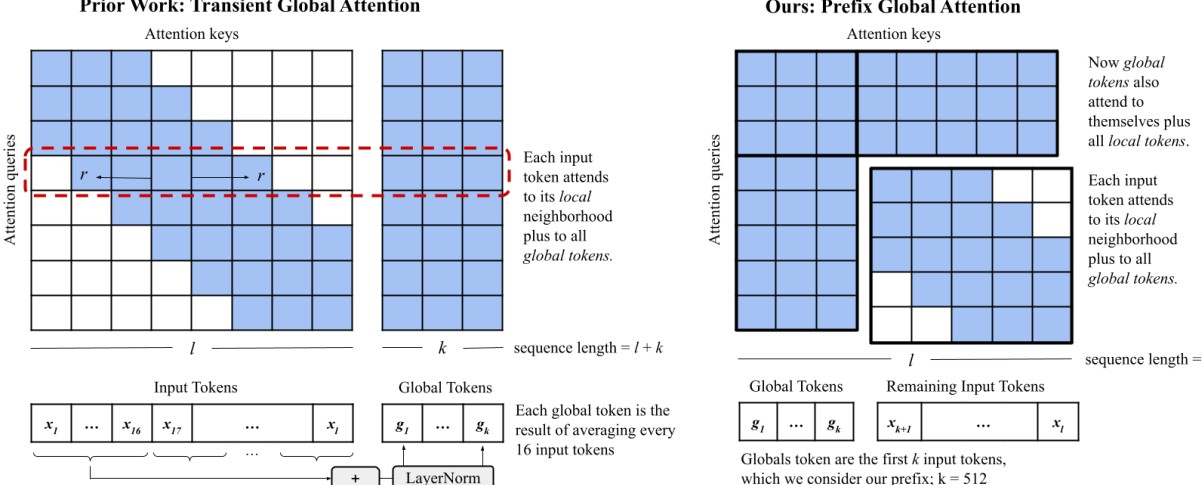

Figure 2: Local-global attention schemes. On the left we show Transient Global (TGlobal), which has local to local and local to global attention (Guo et al., 2022). We propose the new Prefix Global attention which additionally has global to global and global to local attention compared to TGlobal. We define global tokens as a fixed-length prefix of the input, unlike TGlobal which defines additional global tokens that are aggregates over the full input sequence.

captions that can be reconstructed. A Wikipedia image can have three caption types (not all are always available): the alt-text, reference, and attribution descriptions. Alt-text serves as a text description for accessibility purposes, the reference description comes directly below the image in the rendered webpage, and the attribution description contains captions unique to the image across all webpages it appears in. Prior work only input the image, attribution description and associated section text because that was all that was available.

## 3 Prefix Global Attention

When structured image-text data is available, we need not treat all images and text equally. With webpages, it may be more sensible to isolate certain parts as more important. *E.g.*, in contextual image captioning, the model should focus on the target image and section it came from, while using the rest of the page as additional context. We can now isolate these inputs with the WikiWeb2M dataset because we have structural metadata signaling where each image and text element are located, as opposed to a bag of images and a single long body of text. *I.e.*, the new structure available in our dataset can both serve as new inputs to the model and enable new attention mechanisms.

We thus propose Prefix Global, a local-global attention, to capitalize on this intuition. A mixture of local and global attention weights provides the means to designate certain inputs as "global" tokens which can specially attend to the rest of the

input sequence, while others only have local attention to a radius of $r$ tokens to the left and right. Not only is it desirable to prioritize more salient image and text content from the input data, but it can also reduce the computational complexity of the attention mechanism. While full attention is performant by allowing all input tokens to attend to each other, it results in a quadratic computational complexity ($O(l^2)$ for a sequence of length $l$).

Figure 2 illustrates our Prefix Global and prior work's Transient Global attention schemes, where in each the $i$th row represents what the $i$th token can attend to. Guo et al. (2022) introduced LongT5 as an adaptation of the T5 model with Transient Global (TGlobal) attention to balance the efficiency of local attention, which allows for much longer input sequences to be held in memory, with the higher performance of full attention. TGlobal resulted in similar or better performance than full attention with much longer sequence lengths, while having a complexity of $O(l(r + k))$ for a variety of text summarization tasks (where $k = \frac{l}{16}$).

In addition to the local attention of TGlobal (see the blue attention diagonal in Figure 2 (left)), "transient" global tokens are defined on the fly per layer. TGlobal defines $k$ globals as the average of every 16 input tokens, which are additionally attended to by all other inputs. As a result, TGlobal has more global tokens as the sequence length increases. In contrast, shown in Figure 2 (right), Prefix Global uses a constant number of global tokens. Specifically, it takes a prefix of the input sequence. This is

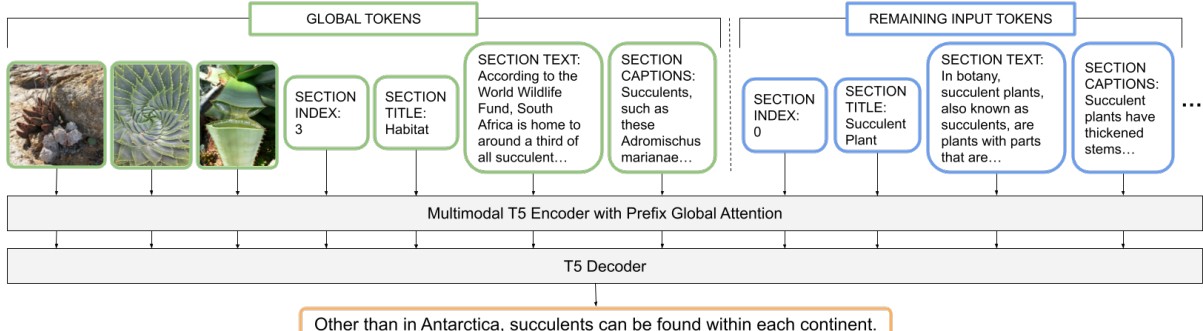

Figure 3: WikiWeb2M section summarization with Prefix Global. With WikiWeb2M, we can now use section structure to separate the most relevant webpage content. The global tokens of Prefix Global (in green) are the first 512 tokens of the target section to be summarized: the first $x$ images of the section, the section index, title, body text, and captions. Then the remaining sections (in blue) from the webpage are input; these have local attention, while the prefix global tokens attend to every other token. We decode the summary (in orange) given the page inputs.

inspired by the leading sentence bias (Kedzie et al., 2018; Xing et al., 2021; Zhu et al., 2021), which shows that earlier content in a body of text is often of greater importance. We define different prefixes for each task in Section 4. While we use section structure to define our prefixes, Prefix Global can use structure from other sources: HTML/the Document Object Model, rendered webpage regions, PDF document layouts, or simply knowing a priori what task inputs are most salient.

Prefix Global has a computational complexity of $O((l - k) \cdot r + k \cdot l)$ for $k$ global tokens, similar to local-global attention schemes ETC (Ainslie et al., 2020), Longformer (Beltagy et al., 2020), and Big-Bird (Zaheer et al., 2020). However, Prefix Global does not require any special pretraining and instead finetunes directly from full attention checkpoints (T5 in our case). This is distinct from LongT5, which also required pretraining with TGlobal attention to be effective. Thus, as we show in Section 5 with Prefix Global's higher performance, it is both a more flexible and performant attention. We also are the first to demonstrate using a local-global attention with multimodal inputs, and further show Prefix Global's ability to be performant in multimodal finetuning from a text-only checkpoint.

## 4 Experiments

We now detail the model variants used for experiments, parameter settings for reproducing our set up, the metrics used for evaluation, and key ablations we perform.

**Model Architectures.** We benchmark with the T5 (Raffel et al., 2020) encoder-decoder framework. T5 takes a sequence of image and text inputs

and we embed images in our input sequence using a frozen ViT model (Dosovitskiy et al., 2021). We note that finetuning ViT may further improve performance. We compare three models defined by different encoder attention schemes: the original T5 which uses full attention, LongT5 with TGlobal attention by Guo et al. (2022) (checkpoints are publicly available), and our Prefix Global attention. We finetune all models from a T5 checkpoint pretrained with full attention on the text-only C4 dataset, and a ViT pretrained on either ImageNet (Deng et al., 2009) or JFT (Hinton et al., 2015).

**Parameter Settings.** We finetune each model for $2^{18}$ steps as done by Raffel et al. (2020) with a 128 batch size. Each model is trained on 16 TPUs, with the base model taking between 24-32 hours to run[4] (varies by task) with an input sequence length of 1024. We do not perform hyperparameter tuning: all models use the Adafactor optimizer with a constant learning rate of 1e-3, an Adafactor offset of 1M to account for pretraining steps, and loss normalizing factor of $2^{18}$. For Prefix Global experiments, the default prefix size $k$ is 512. For both Transient Global and Prefix Global, the local attention neighborhood $r$ is set to 127, as done in LongT5 (Guo et al., 2022).

**Metrics.** For quantitative results, we report BLEU-4 (Papineni et al., 2002), ROUGE-L (Lin, 2004), and CIDEr (Vedantam et al., 2015) metrics from a single run. BLEURT (Pu et al., 2021), CLIP-Score and RefCLIPScore (Hessel et al., 2021) are additionally reported in Appendix C for all results in the main text. We include qualitative results in Appendix B; we perform two qualitative studies

---

[4]Example packing can further improve model efficiency.

to (1) inspect the quality of generated text for all finetuning tasks and (2) discuss when and why images may help the more text-based tasks of page description generation and section summarization.

**Ablations.** We compare each attention at different input lengths. Our sequence length ablations also include experiments where Prefix Global and TGlobal have the same number of global tokens to strictly compare *how* they define global tokens. Then we ablate webpage inputs (section text, titles, structure, images, image captions) and use the best feature combinations for any remaining experiments. We run experiments with different model sizes (B16 or L16 T5 + ViT[5]) for Prefix Global at a 1k input sequence length. Lastly, we verify that WikiWeb2M's new annotations improve performance over prior work. Specifically, we ablate if the target, description, or context sections are input and if sections only from WIT vs. WikiWeb2M are input (since many text and multimodal context sections were not originally kept in the WIT dataset).

### 4.1 Defining Prefix Global Attention Inputs

Each sample's images are always included as part of the input's prefix tokens. We ablated the number of images that contribute to each task's prefix and include ablations in Appendix D.3. We use six images for page description and one image input for section summarization and image captioning.

We describe each task's prefix below. Note that we remove the text that serves as the target summary or caption from our inputs to the model for each task; this ensures there is no model "cheating." *E.g.*, for section summarization, since we utilize the first sentence of a section as its pseudo target summary, we remove it from the inputs to the model.

**Page Description.** We input the images, page URL, page title, and all sections (index, title, text, captions) in their structured page order. In addition to the images, URL, and page title participating in the prefix, we also include all section titles and section first sentences (up to 512 tokens). This outperformed keeping the section titles and text concatenated in order; see Appendix D.1.

**Section Summarization.** The target section to be summarized is prepended to each sample's input sequence. This means the target section's index, title, non-summary text, images, and captions contribute to the global tokens of Prefix Global. Then the page

[5]The base/large T5 model used 220M/770M parameters.

URL, title, and remaining sections follow in order. Figure 3 illustrates how an input sequence is defined with Prefix Global for section summarization.

**Contextual Image Captioning.** Similar to section summarization, the target image and its originating section's content contribute to the prefix tokens (the index, title, text, and non-target captions), followed by the URL, page title, and context sections.

## 5 Results

We now detail experimental results, first evaluating performance and efficiency of each attention type at different sequence lengths. Then, we report input feature, model size, and annotation ablations.

### 5.1 Attention and Sequence Length

**Performance Comparison.** We begin by evaluating performance for each task (page description, section summarization, and contextual image captioning) when training T5 encoders with different attention types and input sequence lengths in Figure 4. Prefix Global always performs better than TGlobal. We include two Prefix Global settings: a fixed $\text{Prefix}_{512}$ which sets 512 input tokens to the prefix (default used for all other experiments), as well as a $\text{Prefix}_{TGlobal}$ which assigns the same number of global tokens as TGlobal. $\text{Prefix}_{TGlobal}$ uses $\frac{l}{16}$ globals, where $l$ is the input sequence length (TGlobal aggregates every 16 input tokens as a global token). This allows us to compare the way both attention mechanisms define global tokens.

Despite TGlobal defining *additional* side inputs as global tokens, it consistently underperforms Prefix Global even with the same number of globals. This confirms that defining a special prefix from the input sequence is better than taking aggregates over the full sequence. In Appendix D.1, we also show that just using the prefix of the in-order page inputs for page description (as opposed to pulling out the section titles and first sentences) performs better than TGlobal. These results collectively show Prefix Global to be preferable to TGlobal. One key takeaway is that separating out more relevant inputs (via structure or other known biases like leading sentence bias) is a good idea.

Full attention and Prefix Global generally have higher performance at longer sequence lengths. It is impressive that Prefix Global scales or maintains performance with larger sequences even when its number of globals is fixed to 512 (*i.e.*, the number of globals is not scaled with respect to input

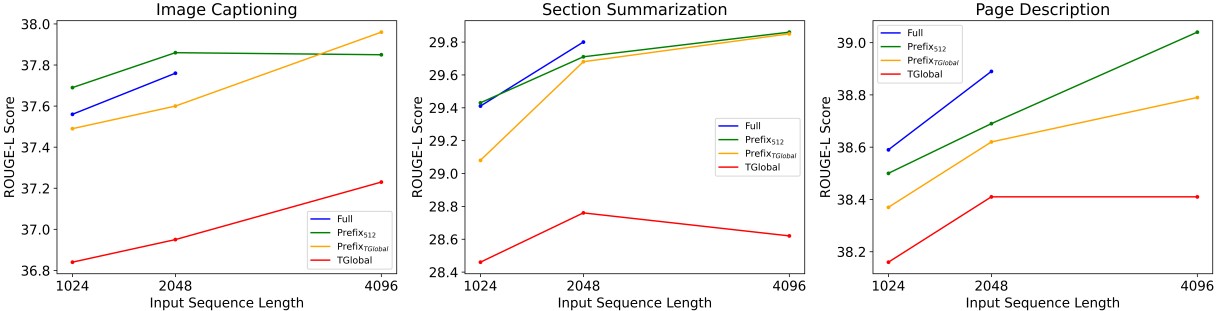

Figure 4: Encoder attention and sequence length experiments. We use Prefix Global, TGlobal, and full attention at 1k, 2k, and 4k sequence lengths. Our experiments verify that Prefix Global is more performant than prior local-global attention TGlobal, and can even be more performant than full attention at long sequence lengths. Note that full attention at the 4k sequence length does not fit into memory. ROUGE-L is plotted.

| Input | Attention Mechanism | | |
|---|---|---|---|
| Length | TGlobal | Prefix Global | Full |
| 1024 | 325,632 | 916,480 | 1,048,576 |
| 2048 | 782,336 | 2,225,152 | 4,194,304 |
| 4096 | 2,088,960 | 4,842,496 | 16,777,216[6] |

Table 4: The approximate number of FLOPs for each attention ignoring the # of attention heads and embedding dimension (both are the same for each attention). As sequence length increases, Prefix Global requires much fewer computations than full attention.

length). On the other hand, while TGlobal scales the number of globals to sequence length, its performance does not consistently scale. *E.g.*, performance plateaus or even drops at 4k input sequence length for page description and section summarization, respectively. This may be because TGlobal defines globals as aggregates over the full input sequence, which could introduce more noise or less semantically rich text at longer sequence lengths.

One anomalous result occurs for image captioning: Prefix Global with 256 globals (Prefix$_{TGlobal}$ at 4k input length) outperforms the 512 variant; as we did not exhaustively ablate the number of global tokens, further performance gains could be reached by optimizing the number of globals per task.

Prefix Global outperforms full attention at all sequence lengths on image captioning, which may be due to the global tokens including the target image and most relevant section content. This should ease the process of learning the most relevant tokens by allowing full attention between the first $k$ target section tokens with the rest of the input sequence, while contextual information from other sections has local attention. For section summarization and

page description, Prefix Global outperforms full attention at the 4k sequence length, while full attention cannot fit in memory. Given that the entire page's content can be useful for generating a page level description, it is sensible that full attention may perform better for smaller sequence lengths as it allows for attention between all input tokens.

**Efficiency Comparison.** Prefix Global can outperform full attention, while only requiring $O((l - k) \cdot r + k \cdot l)$ attention complexity for $k$ global tokens. When implementing the Prefix Global attention, we manually created tensors representing block sparsity to avoid computing the full cross attention. We provide the approximate number of FLOPs for each attention mechanism in Table 4 when ignoring the number of attention heads and embedding dimension. At the 2k input sequence length Prefix Global requires about half the FLOPs of full attention, and experimentally takes about half the time to complete the same experiment with all other settings fixed. The number of FLOPs of Prefix Global at 4k is just over those of full attention at the 2k input length, and is able to fit into memory and maximize performance for each task.

Lastly, the full attention and Prefix Global FLOP difference grows with sequence length. This can sometimes be seen experimentally: performance gaps are larger between full and Prefix Global for page description at 2k vs. 1k (0.20 vs. 0.09).

## 5.2 Feature Ablations

We investigate the role of each input feature with Prefix Global attention and fix sequence length to 1k. Starting with just the text available from web-

---

[6]Full attention will OOM at 4k input length.

[7]Structure features are kept if they were helpful in text-only experiments. *I.e.*, they are included for page description and image captioning, but not for section summarization.

| Feature Inputs | | | | | Page Desc. | | | Section Summ. | | | Image Caption. | | |
|---|---|---|---|---|---|---|---|---|---|---|---|---|---|
| Text | Title | Struct | Caption | Image | B | R | C | B | R | C | B | R | C |
| ✔ | | | | | 13.60 | 37.75 | 77.12 | 9.48 | 28.35 | 65.75 | 9.83 | 33.00 | 133.70 |
| ✔ | ✔ | | | | 13.63 | 37.88 | 77.97 | 9.78 | 29.14 | 68.90 | 9.84 | 33.40 | 135.30 |
| ✔ | ✔ | ✔ | | | **14.07** | 37.96 | 77.88 | 8.70 | 29.24 | 69.19 | 10.15 | 33.38 | 135.10 |
| ✔ | ✔ | | ✔ | | 13.12 | 38.43 | 81.19 | 10.08 | 29.23 | 69.45 | 9.90 | 33.57 | 136.03 |
| ✔ | ✔ | ✔ | ✔ | | 13.22 | 38.38 | 81.38 | 9.51 | 29.22 | 69.24 | 10.03 | 33.69 | 137.07 |
| ✔ | ✔ | ✔[7] | | ✔ | 13.16 | 37.96 | 78.39 | 9.31 | 29.20 | 69.19 | 11.74 | 37.46 | 156.34 |
| ✔ | ✔ | ✔ | ✔ | ✔ | 14.00 | **38.50** | **81.49** | **10.12** | **29.43** | **69.89** | **11.84** | **37.69** | **158.19** |

Table 5: Feature ablations with WikiWeb2M. We ablate over the section body text, title, structure, captions, and images. Utilizing multimodal inputs results in the best performance for all tasks. We report BLEU-4 (B), ROUGE-L (R) and CIDEr (C) metrics.

| Task | Model | ViT Data | Metric | | |
|---|---|---|---|---|---|
| | | | B | R | C |
| Page Desc. | Base | im21k | 14.00 | 38.50 | 81.49 |
| | | JFT | 13.25 | 38.49 | 82.02 |
| | Large | im21k | **14.67** | **39.63** | **88.90** |
| | | JFT | 14.56 | 39.56 | 88.48 |
| Section Summ. | Base | im21k | 10.12 | 29.43 | 69.89 |
| | | JFT | 10.15 | 29.40 | 70.03 |
| | Large | im21k | 11.10 | **30.61** | 76.87 |
| | | JFT | **11.24** | 30.54 | **76.92** |
| Image Cap. | Base | im21k | 11.84 | 37.69 | 158.19 |
| | | JFT | 11.66 | 37.35 | 156.01 |
| | Large | im21k | **12.51** | **38.05** | **162.31** |
| | | JFT | 12.08 | 37.33 | 158.81 |

Table 6: Pretrained model checkpoint ablations. We vary the size of the T5 and ViT models (Base means both T5 and ViT are base-sized models, Large means both are large models) and which image dataset the ViT model was pretrained with (ImageNet or JFT-300M).

page sections, we incrementally add section titles, indices and special tokens defining section structure (the struct column of Table 5), the captions of images within each section, and the images. Each input boosts performance[8] except section structure which has mixed results; for multimodal experiments we include these extra tokens if they helped in the text-only experiments. This may be due to these extra tokens consuming global tokens in the prefix that otherwise could have been more useful.

Images and their captions both improve performance, but result in the highest performance for each task when used in tandem. This illustrates that even when text captions are available, having their visual counterpart is beneficial. In Table 5,

when we include captions for the image captioning task, it refers to *context* captions from other images in the page that never serve as target images. Interestingly, this boosts performance. We suspect contextual captions help the model to learn the style of captions we aim to generate.

### 5.3 Pretrained Checkpoint and Model Size

In Table 6, we perform additional experiments with ViT pretrained on JFT and large T5/ViT models. Unsurprisingly, larger models result in better performance. For page description and section summarization, scaling the model size results in larger performance gains than the impact of any individual feature we ablated. On the other hand, model size has smaller gains for image captioning compared to the impact of our feature ablations; the worst to best performance gap changed by an average of 17.66% for feature ablations and only by 2.43% for model size, where we average the performance delta of BLEU-4, ROUGE-L, and CIDEr.

Preference to ViT representations pretrained on JFT or ImageNet varies by task: section summarization tends to prefer JFT, while page description and image captioning consistently perform best with large ImageNet trained representations.

### 5.4 Comparison to WIT Annotations

The proposed WikiWeb2M is a superset of WIT. For the same set of webpages, we unify all sections into a webpage sample and reintroduce millions of sections and images that were not kept in WIT. Table 7 contains runs when using the original WIT data, the WIT data reprocessed to join the page sections it originally contained, and our WikiWeb2M.

For section summarization, the page description is more important than the other context sections. The page description may be more generally rele-

---
[8] BLEU-4 is less consistent than ROUGE-L and CIDEr.

| Task | Input Section Type | | | Section Source | Metric | | |
|------|:------:|:-----------:|:-------:|:--------------:|:------:|:-------:|:-----:|
| | Target | Description | Context | | BLEU-4 | ROUGE-L | CIDEr |
| Section Summarization | ✔ | | | WikiWeb2M | 8.90 | 27.82 | 60.20 |
| | ✔ | ✔ | | | 9.46 | 28.86 | 66.67 |
| | ✔ | ✔ | ✔ | | **10.12** | **29.43** | **69.89** |
| Image Captioning | ✔ | | | WIT | 10.92 | 36.21 | 148.53 |
| | ✔ | ✔ | | WIT | 11.21 | 36.63 | 150.98 |
| | ✔ | ✔ | ✔ | WIT | 11.45 | 36.88 | 152.69 |
| | ✔ | ✔ | ✔ | WikiWeb2M | **11.84** | **37.69** | **158.19** |

Table 7: Section input ablations. We vary using the target section, page description, and context sections. For image captioning, we also vary whether the sections come from the smaller WIT or our WikiWeb2M superset - we do not run this ablation for section summarization, as it would result in a different number of train/val/test samples. Results show that using all section types and the annotations made newly available with WikiWeb2M improve performance.

vant to all sections, while each section to be summarized contains a distinct topic compared to the context sections from other parts of the webpage. Lastly, we find WikiWeb2M's additional context sections improve captioning performance the most compared to those already available in WIT (comparing the last two rows of Table 7). This confirms the importance of the new annotations in Wiki-Web2M compared to those available in prior work.

## 6 Related Work

Webpage tasks have been studied with text only HTML for web element classification, HTML description generation, and web navigation. Gur et al. (2022) proposed finetuning Large Language Models for these tasks. Reinforcement Learning methods have also trained agents to perform language commands in handcrafted web environments (Gur et al., 2019; Liu et al., 2018; Jia et al., 2019).

Wikipedia has previously been used to develop downstream tasks. For example, WIT (Srinivasan et al., 2021) released image-caption pairs from Wikipedia, in addition to some contextual section text. While WIT does not contain all of the page content, Nguyen et al. (2022) studied contextual image captioning with the available annotations. This is a webpage task and not strictly an image-text problem, as additional section text is included to aid in Wikipedia image captioning, where captions often contain finer-grained, knowledge based information. AToMiC also studied ways to improve multimodal retrieval with Wikipedia by defining more realistic evaluation sets (Yang et al., 2023).

Aghajanyan et al. (2022) proposed CM3, a Transformer with a causally masked pretraining objective. While CM3 relied on pretraining data

from the web containing the images and HTML of a webpage, this dataset was not open sourced. Their results illustrated that rich HTML data could be used to learn representations for tasks such as image generation, image in-filling, and entity disambiguation and linking. This demonstrates that webpage data can generalize to non-webpage tasks, but leaves webpage specific problems unexplored.

To our knowledge there is no open source multimodal webpage data that captures all modalities. C4 was recently extended to a multimodal version, MMC4 (Zhu et al., 2023). However, MMC4 does not retain structure, and instead uses CLIP scores to noisily match images to chunks of text that it could be aligned with. MMC4 has not yet been used for pretraining or downstream applications. In mobile apps, the closest domain to webpages, there are two open source datasets that contain all modalities (text, image, and structure): Rico (Deka et al., 2017) and MoTIF (Burns et al., 2022).

## 7 Conclusion

In this paper we study three generative tasks for multimodal webpage understanding: page description generation, section summarization, and contextual image captioning. To do so, we present the WikiWeb2M dataset, which retains all of the text, images, and structure from more than 2M pages. We propose a new attention, Prefix Global, which outperforms full attention by allowing the most salient text and images to specially attend to all inputs. Extensive ablations on attention mechanism, sequence length, model size and checkpoint, input features and section type reveal the most impactful factors on our benchmark suite and verify using WikiWeb2M to study webpage understanding.

## Limitations

The WikiWeb2M dataset reprocessed the webpages available in WIT. We begin with only the English subset of WIT, while it originally contained 108 languages. Our dataset is limited to English and does not cover the vast multilingual data on Wikipedia. We can extend our dataset to cover all languages in WIT, but acknowledge it is monolingual to date.

For page description generation and section summarization, we use pseudo summaries that are readily available from Wikipedia pages. While this is desirable from a scalability perspective and is practiced in other works, it can limit the evaluation quality of these tasks. However, we did perform a small scale pilot to collect human annotations for the section summarization task in which we asked the annotators if the first sentence sufficed; 94% of the time the majority vote out of five was yes. Pseudo summaries have also been used for other tasks like summarizing instructional videos (Narasimhan et al., 2022).

For the model settings we explore, we did not try all exhaustive combinations of features, attention mechanism, model configuration, and input length. We also only use T5 variants, but note T5 is state-of-the-art for generation style problems. Lastly, we design our set of fine-tuning tasks for generative tasks. Our work currently does not include tasks like webpage taxonomy classification or webpage retrieval, but additional tasks like topic classification could be performed with WikiWeb2M.

## Ethics Statement

While the Internet provides a vast and rich domain to collect data from, it also has potential risks. Wikipedia is a highly curated and monitored knowledge base of articles, but it can be edited by the public, which can create potential quality risks. Additionally, Wikipedia is a largely fact-based domain, where incorrectly summarizing an article could result in misinformation. We hope our dataset can be used as a new resource to improve the accuracy and factual correctness of text generation machine learning models. As we use Wikipedia data, there is no user data nor P.I.I. in the proposed WikiWeb2M dataset. Additionally, we ran analysis to remove a small subset of pages with potentially sensitive topics (*e.g.*, natural disasters, funeral, blood).

## Acknowledgements

This work is supported, in part, by the Google Ph.D. Fellowship program.

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

## A  Additional Dataset Details

### A.1  Dataset Processing

We now provide additional details on the filters and data processing steps used to convert WikiWeb2M into our three downstream task datasets.

For page description, we retain a page from WikiWeb2M if it is not list-heavy and contains at least two sections with image or text content that *do not* contain a list or table. A small subset of Wikipedia pages are essentially lists[9]; we consider pages that

---

[9]For example, https://en.wikipedia.org/wiki/List_of_mammals_of_the_United_States

explicitly have "list_of" in their URL to be list heavy-pages and remove them. We also remove pages with fewer than two rich sections to ensure there is enough content for a page description task to be appropriate.

For a page section to serve as a target section for the task of section summarization, we require it to have at least five sentences, contain neither a table nor list, and not be the root section. We filter out the root because the root (first) section is often the page description, which we choose to keep as a distinct task.

Lastly for image captioning, we follow Nguyen et al. (2022) and use the reference description as the ground truth caption to be generated. However, unlike Nguyen et al. (2022), we do not input the attribution description for the target image to be captioned because it often heavily overlaps with the reference description (the reference description is used as the target text to generate). We further discuss this design choice in Appendix E. Again, we only consider images that were originally in WIT as target images to be captioned to ensure quality captions. We also only keep target images to be images which have a reference description of at least three words.

We additionally note that in WikiWeb2M we release all three types of captions for each image (the alternative-text, reference description, and attribution description), although not all three are always available for each image. The alternative text caption for images is used for accessibility purposes, and future work can focus on generating these descriptions, as opposed to the reference description.

### A.2  Dataset Noise

Our dataset is built from the web, being processed from raw HTML. Noise may exist in our dataset in the formatting of text, *e.g.*, mathematical formulas may have additional formatting text around them. In building our task datasets, the only noise we may introduce to the best of our knowledge is the processing of first sentences. The first sentence is separated from each section for the section summarization pseudo summary. It is also used as part of the global inputs for page description generation.

Specifically, the code used to parse the first sentence may prematurely split a sentence from a period that does not signal the end of the sentence. We did manually inspect samples early on and found this to be rare (*e.g.*, 96/100 random samples were

split correctly). Additionally, the sentences which were split prematurely could still be valid standalone sentences. Our released dataset does include all raw text, so others can reprocess it as they see fit. Figure 5 includes a code snippet for the sentence preprocessor we use and Table 8 illustrates the 4/100 prematurely split sentences from our random sample.

## A.3 Dataset Analysis

We provide a side by side comparison of the fields we open source with the WikiWeb2M dataset compared to those pre-existing in WIT in Table 9. Note that in addition to the new fields we introduce, WikiWeb2M has different data processing which allows for a great number of sections and images to be retained, as seen in Tables 10-12. In the main text, Table 1 provided the aggregate counts over all splits for each section type and the number of images in our dataset versus prior work WIT. Tables 10, 11, and 12 provide the same statistics but now broken down for train, validation, and test sets, respectively.

In Figure 6, two WikiWeb2M dataset samples are visually illustrated. Specifically, the webpages on the topics of Succulent Plant and the Aguas Livres Aqueduct are shown on the left of the figure to visualize the original webpage for illustration purposes, and on the right we show a subset of the fields available in our WikiWeb2M samples for these same pages.

For additional data analysis we provide sequence length details. We include the median, average, 90th percentile, and maximum sequence length values for all input fields that make up a sample's input sequence in the train split. We define sequence length as the number of tokens after preprocessing and SentencePiece tokenization. See Table 13 for the sequence length of the page URL, title, description, section title and text, and image captions available (the alt-text, attribution description, and reference description).

Lastly, we provide aggregate high level statistics over the train split of WikiWeb2M in Table 14. This includes statistics on the number of sections per page (the number of total sections as well as the number of content sections, with the latter referring to sections which contain image or text content) and the number of images per page or section.

## B Qualitative Examples

To qualitatively evaluate our model's ability on our suite of webpage understanding tasks, we include two qualitative analyses. First, we report several random output samples for page description generation, section summarization, and contextual image captioning in Tables 15-17. In Appendix B.1, we discuss our findings from these randomly sampled outputs. Next, in Appendix B.2, we include sample outputs whose metrics improved the most from the text-only model to the multimodal model, to explore why images can be helpful for webpage understanding tasks. In this second setting we investigate samples for page description and section summarization, since these tasks do not obviously require images in the same manner as contextual image captioning.

## B.1 General Qualitative Examples

We start with our first analysis of randomly sampled outputs for all three fine-tuning tasks. Samples are selected from the test set.

### B.1.1 Page Description Generation

Beginning with page description in Table 15, the target and predicted output text are provided for three random pages on the topics of the Horahora Power Station, Hedevig Lund, and Cape Nome.

For the first article on the Horahora Power Station, the predicted output text is quite coherent and well formed, despite containing some inaccurate details that conflict with the content of the webpage. Our model correctly references the date it was opened (1913) and the date the power station was flooded (1947). It also correctly references Lake Karapiro, which was formed and ultimately led to the submerging of the power station. On the other hand, the name of the "Waikato" River was swapped with "Waihi." The model also referred to Horahora as a coal-fired station when it is actually a hydroelectric power station.

Next, for the shorter article on Hedevig Lund, we find the model prediction to be very close to the target page description, although the painter's last names are slightly incorrect. Upon inspecting the page text, it appears the model included additional last names from the painter's parents' names (Ole Wilhelm Erichsen and Abel Marie née Isaachsen). In future work, methods that use pointer networks or direct copying from input text can be used to ameliorate these named entity failures.

```
 1  alphabets = "([A-Za-z])"
 2  prefixes = "(Mr|St|Mrs|Ms|Dr|Prof|Capt|Cpt|Lt|Mt)[.]"
 3  suffixes = "(Inc|Ltd|Jr|Sr|Co)"
 4  starters = "(Mr|Mrs|Ms|Dr|He\s|She\s|It\s|They\s|Their\s|Our\s|We\s|But\s|However\s|That\s|This\s|Wherever)"
 5  acronyms = "([A-Z][.][A-Z][.](?:[A-Z][.])?)"
 6  websites = "[.](com|net|org|io|gov|me|edu)"
 7  digits = "([0-9])"
 8
 9  def PreprocessText(og_text):
10      text = " " + og_text + "  "
11      text = text.replace("\n", " ")
12      text = re.sub(prefixes, "\\1<prd>", text)
13      text = re.sub(websites, "<prd>\\1", text)
14      text = re.sub(digits + "[.]" + digits, "\\1<prd>\\2", text)
15      text = re.sub("\s" + alphabets + "[.] ", " \\1<prd> ", text)
16      text = re.sub(acronyms + " " + starters, "\\1<stop> \\2", text)
17      text = re.sub(alphabets + "[.]" + alphabets + "[.]" + alphabets + "[.]", "\\1<prd>\\2<prd>\\3<prd>", text)
18      text = re.sub(alphabets + "[.]" + alphabets + "[.]", "\\1<prd>\\2<prd>", text)
19      text = re.sub(" " + suffixes + "[.] " + starters, " \\1<stop> \\2", text)
20      text = re.sub(" " + suffixes + "[.]", " \\1<prd>", text)
21      text = re.sub(" " + alphabets + "[.]", " \\1<prd>", text)
22      text = re.sub(r" (\d+)[.](\d+) ", " \\1<prd>\\2 ", text)  # decimal numbers
23
24      if """ in text:
25          text = text.replace(".""", """.")
26      if "\"" in text:
27          text = text.replace(".\"", "\".")
28      if "!" in text:
29          text = text.replace("!\"", "\"!")
30      if "?" in text:
31          text = text.replace("?\"", "\"?")
32      if "e.g." in text:
33          text = text.replace("e.g.", "e<prd>g<prd>")
34      if "i.e." in text:
35          text = text.replace("i.e.", "i<prd>e<prd>")
36      if "..." in text:
37          text = text.replace("...", "<prd><prd><prd>")
38      if "Ph.D" in text:
39          text = text.replace("Ph.D.", "Ph<prd>D<prd>")
40
41      text = text.replace(".", ".<stop>")
42      text = text.replace("?", "?<stop>")
43      text = text.replace("!", "!<stop>")
44      text = text.replace("<prd>", ".")
45      return text
46
47  def GetFirstRestSentences(og_text):
48      """Splits a body of text into its first sentence and the remaining text.
49
50      Code modified from
51      https://stackoverflow.com/questions/4576077/how-can-i-split-a-text-into-sentences
52
53      Args:
54          og_text: The input text string.
55      Returns:
56          first: The first sentence of a body of text.
57          rest: The remaining, rejoined sentences from a body of text that follow the first sentence.
58      """
59
60      text = PreprocessText(og_text)
61      sentences = text.split("<stop>")
62      sentences = sentences[:-1]
63      sentences = [s.strip() for s in sentences]
64
65      first = sentences[0]
66      rest = og_text[len(first)+1:]
67      return first, rest
```

Figure 5: The code used for sentence splitting. This preprocessing is used to separate the first sentence from the rest of the section body for section summarization targets and for page description generation global input tokens.

| Full First Sentence | Processed First Sentence |
|---|---|
| In 2007, Saade was a founding member of What's Up!, a Swedish boy band which included Robin Stjernberg, Ludwig "hejLudde" Keijser and Johan Yngvesson. | In 2007, Saade was a founding member of What's Up! |
| Unicamp offers over one thousand extension programs to the community, with different levels of minimum requirements (high school degree, undergraduate degree, etc.) and across all areas of study, focusing mainly on specialization courses and community outreach. | Unicamp offers over one thousand extension programs to the community, with different levels of minimum requirements (high school degree, undergraduate degree, etc. |
| The proposed biosynthesis of ascofuranone was reported by Kita et al., as well as by Abe et al. | The proposed biosynthesis of ascofuranone was reported by Kita et al. |
| 1988 The choir sang the German premiere of Joseph Jongen's Mass for choir, brass ensemble and organ, Op. 130, which was not yet in print then, both in the Stiftskirche of Aschaffenburg and in St. Bonifatius. | 1988 The choir sang the German premiere of Joseph Jongen's Mass for choir, brass ensemble and organ, Op. |

Table 8: Failures of our sentence processor. We found 4/100 randomly sampled sections had first sentences which were prematurely split. We share these captions here and find they still are fairly reasonable sentences. The "full" first sentence on the left is determined via manual inspection.

The Cape Nome article is another example with a slightly longer page description (four sentences). This sample strongly illustrates the model's ability to convey a factually accurate and topically relevant page description even when the output text does not entirely contain the same content as the target description. Both the target and predicted descriptions begin with an overall summary of the topic, followed by geographical information. Our model's generated text also provides some historical context that is accurately summarized from the article, which the ground truth description does not. It seems our model attempts to summarize each of the sections on the page to form a coherent page description, which may differ from the target page description on Wikipedia (*i.e.*, the Wikipedia page description need not cover topics from the entire webpage and can vary in style page to page).

### B.1.2 Section Summarization

For the task of section summarization, we include links to the webpage and the target section to be summarized from the article, and the target and predicted text in Table 16. Starting with the historical section on imageboards, we find that the target section summary is slightly more section specific than the predicted summary. *I.e.*, the model generated summary "Futallaby is a free and open-source imageboard script" could be in sections other than the historical section. That being said, the historical section does discuss that Futallaby is freely available, making the model predictions sensible, relevant, and factually correct.

In the second section summarization example on the topic of Jamaica's pirate economy, the target summary discusses Spanish resistance to English occupancy to provide context for the growing pirate economy. However, neither the target nor predicted section summary directly address the pirate economy. The model prediction is mostly accurate with correct references to English occupancy in 1655, but implicitly refers to Port Royal as a fort at the foot of the Blue Mountains, which geographically, is slightly questionable.

The third section summarization sample concerns the science fantasy novel "Thuvia, Maid of Mars," written by Edgar Rice Burroughs. Our trained model correctly references Burroughs finishing a novel by June 1914, but it was Thuvia, Maid of Mars he finished, not the book "Tarzan." The model seems to have confused multiple facts relating to this book: Thuvia, Maid of Mars was the fourth, not third, novel in the Barsoom series and Tarzan was not a part of this novel series (although Burroughs did also write Tarzan).

| WikiWeb2M Field | WIT Field |
| --- | --- |
| page_title | page_title |
| page_url | page_url |
| raw_page_description | context_page_description |
| section_title | section_title |
| section_text | context_section_description |
| section_image_url | image_url |
| section_image_mime_type | image_mime_type |
| section_image_width | original_width |
| section_image_height | original_height |
| section_image_raw_ref_desc | caption_reference_description |
| section_image_raw_attr_desc | caption_attribution_description |
| section_image_alt_text_desc | caption_alt_text_description |
| clean_page_description | – |
| section_image_clean_ref_desc | – |
| section_image_clean_attr_desc | – |
| section_image_captions | – |
| section_index | – |
| section_raw_1st_sentence | – |
| section_clean_1st_sentence | – |
| section_rest_sentence | – |
| section_depth | – |
| section_heading_level | – |
| section_subsection_index | – |
| section_parent_index | – |
| page_contains_images | – |
| section_contains_images | – |
| section_image_in_wit | – |
| split | – |
| is_page_description_sample | – |
| page_content_sections_without_table_list | – |
| section_contains_table_or_list | – |
| is_section_summarization_sample | – |
| is_image_caption_sample | – |

Table 9: Dataset fields in our WikiWeb2M dataset versus the WIT dataset.

### B.1.3 Contextual Image Captioning

Lastly, in Table 17, we provide qualitative examples for contextual image captioning. We include links to the webpage and image (as well as illustrate the image within the table), plus the target and predicted image captions. In the first image from the Longpré-le-Sec commune in France, while the target caption describes the main road in the image, a church is also present at the end of the road. This is confirmed by another image on the webpage which shows the same church. Thus, while our model did not predict the same exact target caption, it is still visually and factually accurate.

The second image is a photo of a painting of a woman. This image has a more generic target caption, and it appears that our model tends to prefer generating detailed captions. As a result, it contains factually inaccurate information, stating the painting itself is of Rabindranath, the son of Maharshi Devendranath Tagore, who developed Santiniketan. Additionally, the generated caption states the mural is at the Ashram Complex in Santiniketan. While the painting is at Santiniketan, it is not confirmed to be at the Ashram Complex given the content of the article; while the article states "It [the Ashram Complex] has beautiful frescoes by Nandalal Bose," it remains ambiguous.

Then, for the third randomly selected image from

| Dataset | # Webpage Sections | | | | | | # Images | |
|---|---|---|---|---|---|---|---|---|
| | Structural | Heading | Text | Image | Both | Total | Unique | Total |
| WIT (En) | - | - | - | 179,769 | 2,562,275 | 2,742,044 | 3,183,132 | 4,456,169 |
| WikiWeb2M | 658,241 | 616,534 | 6,134,086 | 199,165 | 2,911,268 | 10,519,294 | 3,867,277 | 5,340,708 |

Table 10: Comparison of WikiWeb2M and the WIT (Srinivasan et al., 2021) dataset. We report counts here for the train split of WikiWeb2M.

| Dataset | # Webpage Sections | | | | | | # Images | |
|---|---|---|---|---|---|---|---|---|
| | Structural | Heading | Text | Image | Both | Total | Unique | Total |
| WIT (En) | - | - | - | 10,198 | 142,737 | 152,935 | 238,155 | 249,313 |
| WikiWeb2M | 36,410 | 34,274 | 341,310 | 11,276 | 162,381 | 585,651 | 284,975 | 299,057 |

Table 11: Comparison of WikiWeb2M and the WIT (Srinivasan et al., 2021) dataset. We report counts here for the val split of WikiWeb2M.

the article on WWOR-TV, the model's generated caption is quite accurate to the image and also overlaps heavily with the content of the ground truth caption. The only subtle inaccuracy in the predicted text is that it states the TV logo was in use from the early 1970s to early 1980s, when it was actually used until the year 1987, which should be considered the late 1980s.

## B.2 Unimodal to Multimodal Qualitative Examples

We now select a random subset of test set outputs for page description and section summarization for the best performing text-only model and the best performing multimodal (image and text) model. These models are base size T5/ViT models, as that was the model size used to perform feature ablations. Specifically, we select samples which have a higher ROUGE-L score with the multimodal model. For a random subset of 1000 samples, we reverse sort by the change in ROUGE-L between the unimodal and multimodal models, looking to inspect the samples most positively impacted by the inclusion of images. We hope to understand the settings under which images can aid these tasks.

As noted previously, we include this analysis for page description and section summarization, since these tasks may not require images, while image captioning inherently uses the input image. These examples are included in Tables 18 and 19. We did find that some of the most improved samples were an artifact of the text-only model repeating text tokens many times (a common failure of text generation models) and do not include those in our examples.

### B.2.1 Page Description

Starting with page description generation, we include three examples in Table 18. For each page we link to the Wikipedia article and include the target page description, the page description generated by the text-only model (noted as text under the type column), and the page description generated by the multimodal model (noted as multi under the type column) for comparison.

Across these examples, we find one trend: images in the webpage can improve the generated description's *specificity*. In the page description task, we allow up to six images present on the page to be included. Starting with the first example from the webpage on Joan Carling, we see that the page description output from the multimodal model touches upon more topics and specifies details beyond a high-level sentence summary (the latter being more similar to the output from the text-only model). The multimodal model's generated text includes references to the Champions of the Earth Lifetime Achievement Award that was given to Joan Carling as well as greater detail about her relationship with the Philippines and being of the Igorot people. These events and concepts are both captured in the images in the page, which seem to help specify more detail from the page in the generated description.

Similarly, for the second example from the page on Alice Bemis Taylor, the text-only model generated a brief and high level summary about her. On the other hand, the images on the webpage portray numerous important locations (either her childhood home or community spaces she actively participated in or founded such as the Colorado Springs Arts Center and Day Nursery). As a result of in-

| Dataset | # Webpage Sections | | | | | | # Images | |
|---|---|---|---|---|---|---|---|---|
| | Structural | Heading | Text | Image | Both | Total | Unique | Total |
| WIT (En) | - | - | - | 9,905 | 142,917 | 152,822 | 238,924 | 250,353 |
| WikiWeb2M | 36,743 | 35,568 | 342,554 | 11,082 | 162,605 | 588,552 | 286,390 | 300,666 |

Table 12: Comparison of WikiWeb2M and the WIT (Srinivasan et al., 2021) dataset. We report counts here for the test split of WikiWeb2M.

| Sequence Length | Med | Avg | Max | $P_{90}$ |
|---|---|---|---|---|
| Page URL | 16 | 17.40 | 89 | 23 |
| Page Title | 5 | 5.20 | 42 | 8 |
| Page Description | 110 | 122.27 | 695 | 289 |
| Section Title | 3 | 3.64 | 176 | 7 |
| Section Text | 119 | 212.87 | 62,418 | 523 |
| Image Alt-Text | 4 | 5.58 | 1,433 | 14 |
| Image Attribution | 17 | 29.76 | 30,121 | 59 |
| Image Reference | 9 | 8.20 | 3,640 | 27 |

Table 13: Sequence length statistics for various Wiki-Web2M data fields. Sequence length is defined by the number of SentencePiece Tokens. We report the median, average, maximum, and 90th percentile sequence length values for the train split.

| Statistic | Med | Avg | Max | $P_{90}$ |
|---|---|---|---|---|
| Total Sections / Page | 4 | 5.83 | 987 | 12 |
| Content Sections / Page | 4 | 5.13 | 939 | 11 |
| Images / Page | 1 | 2.96 | 1,679 | 6 |
| Images / Section | 0 | 0.51 | 653 | 1 |

Table 14: Analysis for the number of sections and images per page or section in the train split of the Wiki-Web2M dataset. Here we consider content sections as a section with text or image content within it (as in not only containing lists, tables, or heading only).

cluding these images, the generated description now included more specific information regarding these places. By including these images, their corresponding captions and related textual concepts in the input sequence are more greatly attended to. This may be a byproduct of images always serving as global tokens with our Prefix Global attention mechanism.

Lastly, we include the descriptions generated for the webpage on the topic of the Franco-German Parliamentary Assembly. Again this example demonstrates that images can help maintain the correct level of specificity for the page description. In this case, the images help refocus the description to be more high-level than the description generated from the text model, which includes too much detail (unlike the prior examples which tended towards too little). The two images on this page are centered around the respective meeting locations of the parliamentary in France and Germany, which helps focus the topic of the page description.

### B.2.2 Section Summarization

In Table 19 we now include section summarization examples for several pages and include links to the webpage and specific section to be summarized. Slightly different from the trend found for page description generation, we see that images can improve the *topical relevance* of the generated section summary.

For example, with the webpage on Johann Joachim Quantz, the target section summary for the section on the Court of Frederick discusses how Quantz joined the court as a flute teacher to Frederick II. The image in this section illustrates Frederick the Great playing flute, which directs the model to focus on this instrument. The text-only model fails to mention anything about instruments in the summary, only discussing interpersonal relationships. On the other hand, the multimodal model mentions both the flute and specifically Frederick playing the flute, which is depicted in the included image and also explains the reason why Quantz joined the court.

Next, the section on the 1800s history with respect to the St. Peter Catholic Church (in Montgomery, Alabama) continues to illustrate how images can improve topic relevance. While the text model references the year the church was founded, the multimodal model references the location, which the target summary also describes (although they reference the location of the church at different degrees of granularity). If a specific section does not have images, an image is included from the images in the chronological order they appear on the webpage. This means that for this section, the included image is the first image on the page - a map of Alabama showing the location of the St. Peter Catholic Church.

A third example can be found with the P&T

| Webpage | Target Text | Predicted Text |
|---|---|---|
| Horahora Power Station | Horahora Power Station was an early hydro-electric power station on the Waikato River in New Zealand. It was the country's first large-scale power station, completed in 1913. Initially built to service a gold mine, the power station was expanded to supply a significant part of the North Island. The power station remained in use until was submerged by Lake Karapiro, which was formed to supply the larger Karapiro Power Station. | Horahora Power Station was a coal-fired power station located on the banks of the Waihi River in the Waikato region of New Zealand. It was built in 1913 and flooded in 1947 when Lake Karapiro was formed behind the Karapiro Power Station dam. |
| Hedevig Lund | Hedevig Thorine Christine Erichsen Lund was a Norwegian painter. | Hedevig Lund née Erichsen was a Norwegian painter. |
| Cape Nome | Cape Nome is a headland on the Seward Peninsula in the U.S. state of Alaska. It is situated on the northern shore of Norton Sound, 15 miles to the east of Nome also on Norton Sound. It is delimited by the Norton Sound to the south, Hastings Creek on the west, a lagoon on the east and an estuary formed by the Flambeau River and the Eldorado River. From the sea shore, Cape Nome extends inland by about 4 miles, connected by road with Nome. | Cape Nome is a headland on the Seward Peninsula of Alaska, United States. It is located 129 miles to the south east of the Bering Strait. The Cape Nome mining precinct was established in 1898 immediately after the discovery of gold at Golovnin Bay. |

Table 15: Page description qualitative examples. We include three random samples from the page description test set and compare the target page description and predicted model output text.

Group article and 1900s-1950s Growth and Expansion section. The target section summary discusses the opening of the Shanghai office. The summary generated from the text-only model discusses the P&T Group expanding its business, which while correct, does not discuss the connection between opening new offices and expanding business. On the other hand, with the help of the HSBC Shanghai office image input to the multimodal, the multimodal model generated a description which explicitly mentioned the office opening. However, due to both Shanghai and Hong Kong locations being mentioned in the section, the model confuses the two in the generated section summary.

## C  Additional Metrics

For results reported in the main paper, we additionally report BLEURT, CLIPScore, and RefCLIP-Score (the latter two are only relevant for contextual image captioning). See Tables 20, 21, 22. All results trends stay the same except that BLEURT is insensitive to most of the feature ablation results for section summarization and page description

generation.

## D  Additional Model Ablations

We now provide additional experiments that could not fit into the main text, which include data processing ablations, additional feature ablations, and further comparisons to prior work.

### D.1  Page Description

We first performed ablations on the data processing of WikiWeb2M for the page description generation task dataset. Specifically, we tried varying the number of content sections required for a particular page to be retained. See Table 23 for comparison of when we required two vs. three vs. four sections to contain image or text content without a table or list. We found the added samples improve performance consistently (i.e., the most performant setting is when the number of content sections required per page is set to two).

---

[10]Structure features are kept if they were helpful in text-only experiments. I.e., they are included for page description and image captioning, but not for section summarization.

| Webpage | Section | Target Text | Predicted Text |
|---|---|---|---|
| Imageboard | Historical | Futallaby is a PHP script based on the Futaba script from Futaba Channel. | Futallaby is a free and open-source imageboard script. |
| Colony of Jamaica | Jamaica's Pirate Economy | Spanish resistance continued for some years after the English conquest, in some cases with the help of the Jamaican Maroons, but Spain never succeeded in retaking the island. | The English occupied the island of Jamaica in 1655, establishing a fort at the foot of the Blue Mountains. |
| Thuvia, Maid of Mars | Background | Burroughs began writing Thuvia, Maid of Mars, in April 1914, at the time describing it as a 'Carthoris' story. | By June 1914, Burroughs had completed the third novel in the Barsoom series, Tarzan. |

Table 16: Section summarization qualitative examples. We include three random samples from the section summarization test set and compare the target section summary and predicted model output text.

We also allow for text only pages to be kept in the dataset, as there are a small subset (roughly 2% of pages) that do not have any images after our processing. This could be due to the Wikipedia pages changing since the original WIT dataset was released, or because we only allow JPEG and PNG images while WIT contained some other image types like SVG. We include additional ablations in Table 24 showing the effect of including or not including these unimodal pages; their effect is minimal given how few there are in the dataset.

In Table 25, we show ablations for our prefix design with the page description generation task. Including the section titles and first sentences of each section in the prefix as global tokens improved performance for a majority of metrics, and we kept this set up for the rest of our experiments. We note that even when not using a specially designed prefix (*i.e.*, flattening the section inputs and allowing the first 512 tokens to serve in the prefix, not separating out section titles or first sentences), the Prefix Global attention mechanism still outperforms Transient Global. This follows the principal from leading sentence bias that earlier information in the input text is more important. Thus, if you have a priori knowledge that a particular part of the input is more important than others, separating it into the prefix of our attention mechanism can be effective.

### D.2 Contextual Image Captioning

As image captioning inherently requires images, we performed additional feature ablations on the text features while always including the image (see rows 5-9 in Table 26). We verify in row 5 that when inputting only the image and no contextual text, it is incredibly difficult to generate strong captions for these images which contain a lot of fine-grained information. However, in support of the importance of having *both* images and text inputs, we find that for every text-only to multimodal comparison (where all features are the same except images are included in the latter), the multimodal setting always results in substantial performance gains. For example, quantitatively comparing row 1 and row 6 in Table 26, where either only the section text is input versus the section text and image to be captioned are input, the performance differences are: BLEU-4 9.83 vs. 11.27, ROUGE-L 33.00 vs. 36.90, and CIDEr 133.70 vs. 153.44.

Again, this differs from the findings of Nguyen et al. (2022); their experimental design likely minimized the impact of images because they also feed in the attribution description as a textual input, which often is quite similar to the target caption. As a result, the model can "cheat" and utilize the attribution description while not relying on the visual input. We have more discussion regarding this in Appendix E.

### D.3 All Tasks

For each task in our WikiWeb2M suite, we also ablated the number of images input to the model. These additional ablations are shown in Table 27. Results in the main text use the 90th percentile number of images per page, six, for page description generation, and only one for section summarization

and image captioning. Here we also try the average value for number of images per page which is three. We include these ablations for contextual image captioning as well, as we were curious whether having contextual (non-target) images input to the model would help at all. Ultimately it sometimes hurt performance, likely adding noise and making it more challenging for the model to discern which input image was the target image to be captioned. We only used the single target image as input for the rest of our experiments.

## E  Contextual Image Captioning Task Design

We now provide additional discussion on the task of contextual image captioning and how our input design differs from prior work. Nguyen et al. (2022) recently introduced the task of contextual image captioning. We found our metrics were lower than those they reported for the task and investigated the causes. We ran additional experiments for contextual image captioning with the exact same sample inputs as Nguyen et al. (2022). Specifically, we tried only using the page description, target image, target image's section text, and the attribution description as a sample's inputs.

By including the attribution description (which often heavily overlaps with the target caption to be generated), our performance is much higher, nearly matching prior work even when using different data splits (the prior work's dataset splits are not released). We report these reproduced results for our splits in Table 28. As discussed earlier, for our contextual image captioning task, we chose not to input the attribution description of an image given how much overlap it has with the target caption (the reference description). In terms of other experimental differences, we also use ViT (Dosovitskiy et al., 2021) image representations while prior work used ResNet-152 (He et al., 2016), although both were pretrained on ImageNet (Deng et al., 2009).

## F  Section Summarization Pseudo Summaries

We were motivated to study the task of section summarization as a subproblem of Webpage Story Generation, which is the task of converting a webpage to an Instagram Story-like format. It consists of one multimodal Story page or slide per section, containing a section summary and paired image (from the same webpage). Our section summarization task is a subpart of this problem and we proposed an improvement over the News, text-only CNN/DailyMail PEGASUS model used to generate summaries in the prior Wiki2Story work by Nkemelu et al. (2023). Specifically, our formulation of multimodal section summarization is desirable so that we can also take images as contextual input, as the goal is to generate multimodal content for a user to consume on the topic of a particular webpage (in this case, a Wikipedia article).

Originally, we attempted to collect human written section summaries ourselves. But when running an initial data collection pilot we found that when explaining the intended application of webpage stories, a majority of the annotators deemed the first sentence to almost always (94% of the time) be a good enough pseudo summary. In the other cases when a majority of annotators voted otherwise, we found the annotation quality was too poor to use the collected written summaries. It proved very difficult to collect free form summaries of Wikipedia content. For both of these reasons we continued modeling section summarization with our pseudo summaries (the first sentence of the section).

To perform this data collection pilot, we used an internal crowd sourcing platform to hire seven crowd workers. They were located in India and paid hourly wages competitive for their locale. They have standard rights as contractors. The last four pages of our paper include a PDF of our instructions to annotators. We also tried to collect labels for well suited images for each section but ultimately did not use these annotations.

**Rendered Webpage**                  **WikiWeb2M Sample Fields**

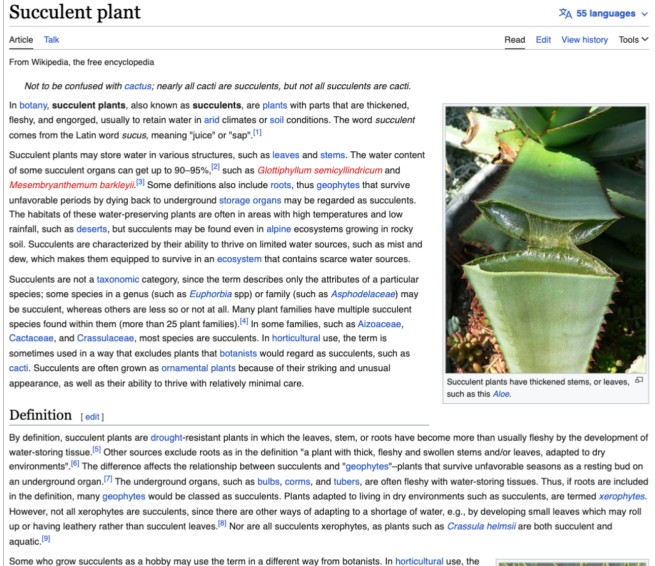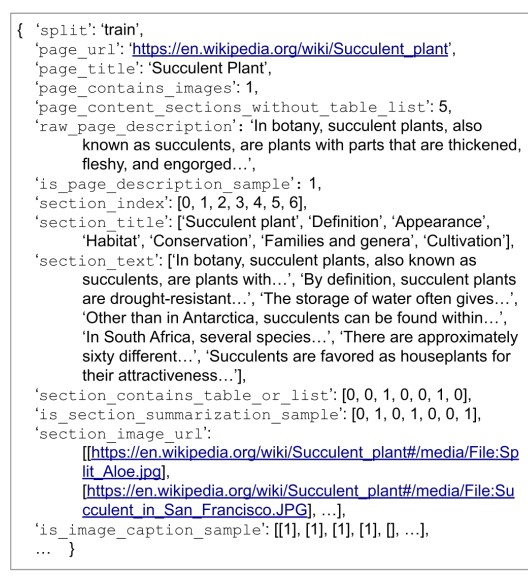

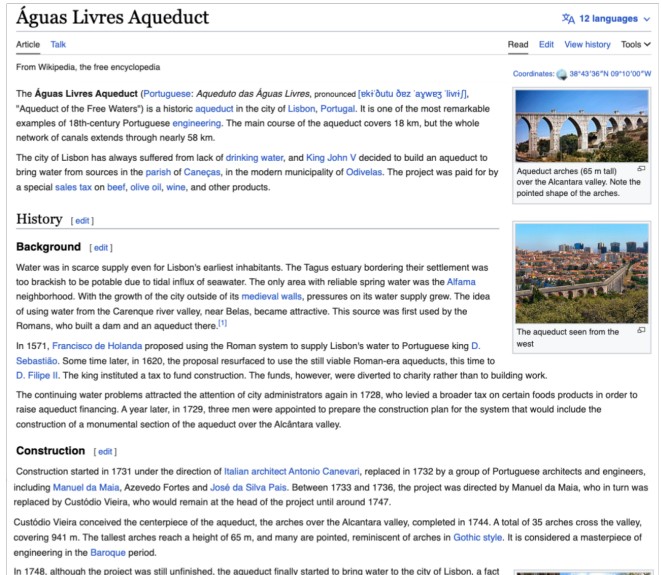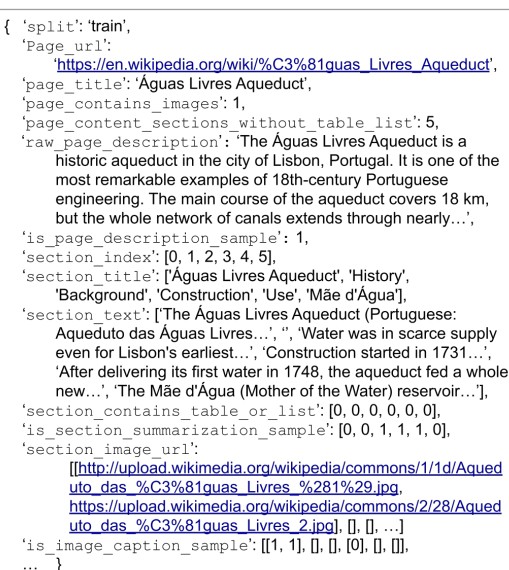

Figure 6: Example samples from WikiWeb2M. Here we illustrate two Wikipedia articles: Succulent Plant and Aguas Livres Aqueduct. The rendered webpage is illustrated on the left. On the right, a sample subset of fields are shown; these are stored in TensorFlow Sequence Example format.

| Webpage | Image | Target Text | Predicted Text |
|---|---|---|---|
| Longpré -le-Sec | 
Link to Image | The main road in Longpré-le-Sec. | The church in Longpré-le-Sec. |
| Santiniketan | 
Link to Image | Mural by Nandalal Bose. | A painting of Rabindranath at the Santiniketan Ashram. |
| WWOR-TV | 
Link to Image | 1971 WOR-TV I.D. slide. This 'dotted 9' logo was used from 1970 to 1987. | WOR-TV logo, used from the early 1970s until the early 1980s. |

Table 17: Contextual image captioning examples. We include three random samples from the image captioning test set and compare the target image caption and predicted model output text.

| Webpage | Type | Text |
|---------|------|------|
| Joan Carling | Target | Joan Carling is an indigenous Filipino human rights activist and environmentalist who has defended the rights of native and marginalized peoples for over two decades. She has served as Secretary General of the Asia Indigenous Peoples Pact and has chaired the Cordillera People's Alliance in the Philippines. Carling has also contributed to the United Nations Framework Convention on Climate Change and REDD+ activities and has served as a member of the United Nations Permanent Forum on Indigenous Issues. In September 2018 she received the Champions of the Earth Lifetime Achievement Award from the United Nations Environment Programme in recognition of her work as an environmentalist and a defender of human rights. |
| | Text | Joan Carling is a Filipino environmentalist and human rights activist. [ROUGE-L 13.11] |
| | Multi | Joan Carling is a Filipino indigenous human rights activist and a member of the Igorot people. She is the former Secretary General of the Cordillera Peoples Alliance, a national Indigenous Peoples' organization. She was the Co-convener of the Indigenous Peoples Major Group on the Sustainable Development Goals. In 2018, she was designated a terrorist by the Philippine authorities for an alleged connection with the Communist Party of the Philippines and the New People's Army. In September 2018, she received the Champions of the Earth Lifetime Achievement Award from the United Nations Environment Programme. [ROUGE-L 39.61] |
| Alice Bemis Taylor | Target | Alice Bemis Taylor was a philanthropist and was inducted into the Colorado Women's Hall of Fame in 2010. For her significant contributions to Colorado College, Colorado Springs Fine Arts Center and the Colorado Springs Day Nursery and other organizations, she was named "Lady Bountiful" by the press. |
| | Text | Alice Bemis Taylor was an American philanthropist and philanthropist. [ROUGE-L 21.05] |
| | Multi | Alice Cogswell Bemis Taylor was an American philanthropist and philanthropist. She was the founder of the Colorado Springs Day Nursery, the Colorado Springs Fine Arts Center, the Child Guidance Clinic and the Colorado Springs Community Chest. She was inducted into the Colorado Women's Hall of Fame in 2010. [ROUGE-L 45.36] |
| Franco-German Parliamentary Assembly | Target | The Franco-German Parliamentary Assembly is a joint body of the German Bundestag and the French National Assembly formed to enable cooperation between both houses. |
| | Text | The Franco-German Parliamentary Assembly is an inter-parliamentary organisation between the French and German parliaments. It was established in March 2019 following the Aachen Treaty, which was signed on 22 January 2019 by Angela Merkel and Emmanuel Macron. [ROUGE-L 28.13] |
| | Multi | The Franco-German Parliamentary Assembly is a joint parliamentary assembly of the French National Assembly and the German Bundestag. [ROUGE-L 59.09] |

Table 18: Page description qualitative examples. We include random samples from the page description test set that had the largest metric improvement with images included, and compare the target page description and predicted model output text.

| Webpage | Section | Type | Text |
|---|---|---|---|
| Johann Joachim Quantz | Court of Frederick | Target | When Frederick II became King of Prussia in 1740, Quantz finally accepted a position as flute teacher, flute maker and composer. |
| | | Text | Quantz was a friend of Frederick the Great, who was a close friend of his father. [ROUGE-L 10.81] |
| | | Multi | Quantz was a friend of Frederick the Great, who was a great lover of the flute. [ROUGE-L 16.22] |
| St. Peter Catholic Church | 1800s | Target | In 1833, arrangements were made to build the church on a property, donated by Edward Hanrick, on the corner of Lawrence Street and Adams Avenue. |
| | | Text | St. Peter Catholic Church was founded in 1834, but had no resident pastor until 1850. [ROUGE-L 5.00] |
| | | Multi | St. Peter Catholic Church is the third oldest Catholic church in Montgomery, Alabama. [ROUGE-L 10.53] |
| P&T Group | 1900s -1950s: Growth and Expansion | Target | In 1920s, the Shanghai office was opened. |
| | | Text | In the early 1900s, the firm expanded its business.[ROUGE-L 25.00] |
| | | Multi | In 1905, the Hong Kong office was opened. [ROUGE-L 66.67] |

Table 19: Section summarization qualitative examples. We include random samples from the section summarization test set which had the largest metric improvement and compare the target section summary and predicted model output text.

| Feature Inputs | | | | | Page Desc. | Section Summ. | Image Caption. | | |
|---|---|---|---|---|---|---|---|---|---|
| Text | Title | Struct | Caption | Image | BLEURT | BLEURT | BLEURT | CLIPScore | Ref CLIPScore |
| ✔ | | | | | **0.51** | 0.43 | 0.36 | 0.6850 | 0.7146 |
| ✔ | ✔ | | | | **0.51** | 0.44 | 0.36 | 0.6845 | 0.7153 |
| ✔ | ✔ | ✔ | | | **0.51** | **0.45** | 0.37 | 0.6865 | 0.7166 |
| ✔ | ✔ | | ✔ | | **0.51** | **0.45** | 0.37 | 0.6851 | 0.7154 |
| ✔ | ✔ | ✔ | ✔ | | **0.51** | **0.45** | 0.37 | 0.6878 | 0.7177 |
| ✔ | ✔ | ✔[10] | | ✔ | **0.51** | **0.45** | **0.41** | **0.7340** | 0.7575 |
| ✔ | ✔ | | ✔ | ✔ | **0.51** | **0.45** | **0.41** | 0.7329 | **0.7576** |

Table 20: Feature ablations with WikiWeb2M. We ablate over the section body text, title, structure, captions, and images. We report BLEURT, CLIPScore, and RefCLIPScore metrics for the results in Table 5.

| Task | Input Section Type | | | Section Source | Metric | | |
|---|---|---|---|---|---|---|---|
| | Target | Description | Context | | BLEURT | CLIPScore | RefCLIPScore |
| Section Summarization | ✔ | | | WikiWeb2M | 0.43 | – | – |
| | ✔ | ✔ | | | 0.44 | – | – |
| | ✔ | ✔ | ✔ | | **0.45** | – | – |
| Image Captioning | ✔ | | | WIT | 0.40 | 0.7287 | 0.7527 |
| | ✔ | ✔ | | WIT | 0.40 | 0.7307 | 0.7537 |
| | ✔ | ✔ | ✔ | WIT | 0.40 | 0.7325 | 0.7558 |
| | ✔ | ✔ | ✔ | WikiWeb2M | **0.41** | **0.7329** | **0.7576** |

Table 21: Section input ablations. We try using only the target section, both the target section and page description and/or context section(s), and vary if the sections come from the smaller WIT or our WikiWeb2M superset. We report BLEURT, CLIPScore, and RefCLIPScore metrics for the results in Table 7.

| Task | Model | ViT Data | Metric | | |
|------|-------|----------|--------|--|--|
| | | | BLEURT | CLIPScore | RefCLIPScore |
| Page Description | Base | im21k | 0.51 | – | – |
| | | JFT | 0.51 | – | – |
| | Large | im21k | **0.52** | – | – |
| | | JFT | **0.52** | – | – |
| Section Summarization | Base | im21k | 0.45 | – | – |
| | | JFT | 0.45 | – | – |
| | Large | im21k | **0.46** | – | – |
| | | JFT | **0.46** | – | – |
| Image Captioning | Base | im21k | **0.41** | 0.7329 | 0.7576 |
| | | JFT | 0.40 | 0.7291 | 0.7534 |
| | Large | im21k | **0.41** | **0.7374** | **0.7611** |
| | | JFT | **0.41** | 0.7263 | 0.7527 |

Table 22: Pretrained model checkpoint ablations. We report BLEURT, CLIPScore, and RefCLIPScore metrics for the results in Table 6.

| # Content Section Filter Threshold | | Metric | | | | |
|------|------|--------|---------|---------|---------|-------|
| Train | Test | BLEU-4 | ROUGE-1 | ROUGE-2 | ROUGE-L | CIDEr |
| 2 | 2 | **13.79** | **45.43** | **27.72** | **38.34** | **81.16** |
| 3 | 2 | 12.38 | 44.35 | 27.05 | 37.63 | 75.35 |
| 4 | 2 | 12.79 | 44.10 | 26.08 | 36.91 | 69.52 |
| 2 | 3 | **13.42** | **44.31** | **26.07** | **36.64** | **69.69** |
| 3 | 3 | 12.32 | 43.52 | 25.81 | 36.28 | 67.56 |
| 4 | 3 | 12.77 | 43.56 | 25.13 | 35.82 | 64.04 |
| 2 | 4 | **12.98** | **43.11** | **24.45** | **34.90** | **59.69** |
| 3 | 4 | 12.01 | 42.41 | 24.28 | 34.63 | 58.03 |
| 4 | 4 | 12.54 | 42.65 | 23.92 | 34.41 | 57.19 |

Table 23: Page description performance across different filtering thresholds. We change the threshold for how many rich content sections a page must have to be included in our page description generation task dataset.

| Task | Sample Modalities | | Metric | | | | |
|------|-------|------|--------|---------|---------|---------|-------|
| | Train | Test | BLEU-4 | ROUGE-1 | ROUGE-2 | ROUGE-L | CIDEr |
| Page Description | Multimodal | Combined | 12.27 | 44.75 | 27.71 | 38.16 | 80.07 |
| | Combined | Combined | **13.79** | **45.43** | **27.72** | **38.34** | **81.16** |
| | Multimodal | Multimodal | 12.30 | 44.67 | 27.58 | 38.03 | 79.00 |
| | Combined | Multimodal | 13.77 | 45.30 | 27.55 | 38.16 | 79.77 |
| Section Summarization | Multimodal | Combined | 9.83 | **34.96** | **15.18** | **29.64** | 70.82 |
| | Combined | Combined | **9.93** | 34.86 | 15.16 | 29.53 | **70.86** |
| | Multimodal | Multimodal | 9.74 | 34.89 | 15.10 | 29.56 | 70.22 |
| | Combined | Multimodal | 9.84 | 34.78 | 15.06 | 29.43 | 70.17 |

Table 24: Comparison of only retaining multimodal page samples versus also allowing for text only pages (combined).

| Prefix Inputs | | | | | | Metric | | |
| --- | --- | --- | --- | --- | --- | --- | --- | --- |
| Images | Page URL | Page Title | Section Title | Section 1st Sentence | All Section Content | BLEU-4 | ROUGE-L | CIDEr |
| ✔ | ✔ | ✔ | | | ✔ | **13.79** | 38.34 | 81.16 |
| ✔ | ✔ | ✔ | ✔ | | | 13.00 | 38.33 | 81.02 |
| ✔ | ✔ | ✔ | ✔ | ✔ | | 12.59 | **38.47** | **81.68** |

Table 25: Prefix Global prefix ablations for page description generation. The input column "all section content" refers to when all section indices, titles, body text, and captions are concatenated in order and then tokens contribute to the prefix up to the first 512 tokens.

| Feature Inputs | | | | | Metric | | |
| --- | --- | --- | --- | --- | --- | --- | --- |
| Text | Title | Struct | Caption | Image | BLEU-4 | ROUGE-L | CIDEr |
| ✔ | | | | | 9.83 | 33.00 | 133.70 |
| ✔ | ✔ | | | | 9.84 | 33.40 | 135.30 |
| ✔ | ✔ | ✔ | | | 10.15 | 33.38 | 135.10 |
| ✔ | ✔ | ✔ | ✔ | | 10.03 | 33.69 | 137.07 |
| | | | | ✔ | 3.18 | 14.55 | 17.43 |
| ✔ | | | | ✔ | 11.27 | 36.90 | 153.44 |
| ✔ | ✔ | | | ✔ | 11.64 | 37.39 | 156.27 |
| ✔ | ✔ | ✔ | | ✔ | 11.74 | 37.46 | 156.34 |
| ✔ | ✔ | ✔ | ✔ | ✔ | **11.84** | **37.69** | **158.19** |

Table 26: Additional feature ablations with WikiWeb2M for contextual image captioning. We ablate over the section body text, title, structure, captions, and images. We report BLEU-4, ROUGE-L and CIDEr metrics. We include rows already reported in the main text for ease of side by side comparison across all feature ablations.

| Task | Number of Input Images | Metric | | |
| --- | --- | --- | --- | --- |
| | | BLEU-4 | ROUGE-L | CIDEr |
| Page Description | 1 | 13.44 | **38.52** | 81.52 |
| | 3 | 13.55 | 38.46 | **82.00** |
| | 6 | **14.00** | 38.50 | 81.49 |
| Section Summarization | 1 | **10.12** | **29.43** | 69.89 |
| | 3 | 10.10 | 29.35 | **70.29** |
| | 6 | 9.67 | 29.37 | **70.29** |
| Image Captioning | 1 | 11.84 | **37.69** | **158.19** |
| | 3 | **11.92** | 37.41 | 157.27 |
| | 6 | 11.84 | 37.45 | 157.20 |

Table 27: Ablations varying the number of input images per task.

| Contextual Image Captioning | Split | Image Input | Text Input | | | | Metric | | |
| --- | --- | --- | --- | --- | --- | --- | --- | --- | --- |
| | | | Desc. | Target Section | Context Section | Attribution Desc. | B | R | C |
| Nguyen *et al*. | Unknown | ResNet | ✔ | ✔ | | ✔ | 23.83 | 48.80 | 276.60 |
| Ours | WikiWeb2M | ViT | ✔ | ✔ | ✔ | | 11.84 | 37.69 | 158.19 |
| | | | ✔ | ✔ | | ✔ | 25.20 | 50.01 | 242.50 |

Table 28: Experimental results when reproducing the task set up of Nguyen et al. (2022). We do not use the same dataset splits since they were not released in prior work. Here our set up uses the same sample inputs as prior work, unlike our result in the main text which does not input the attribution description.

**INSTRUCTIONS FOR ANNOTATORS**

We are collecting annotations for a task called "Wiki2Story." Wiki2Story has so far been defined as a data conversion process from Wikipedia webpages to Wikipedia "stories." The goal is to turn a Wikipedia webpage into an Instagram-like story which contains one story page per Wikipedia section. Each story page includes a section summary and paired image; see the below examples. We provide two examples of what an Introduction and History story page may look like for these sections of the Apple Wikipedia article.

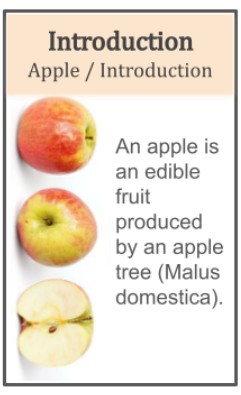
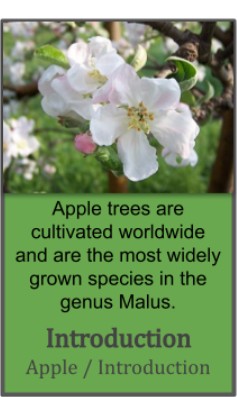
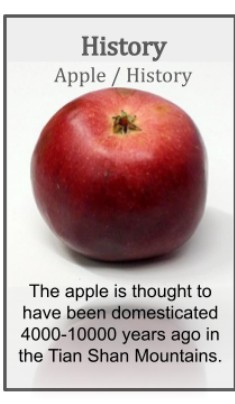
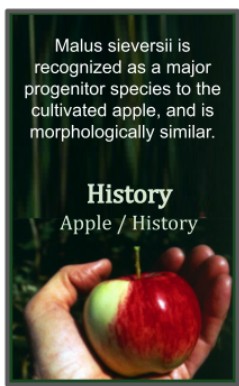

We want to obtain annotations of these section summaries and have you select the most appropriate image to pair with it. In particular, the section summary should be a **highlight** of the section content. The highlight should: be self contained, condense the section's factual information into a sentence of ideally **fewer than 30 words,** and retain enough detail for the reader to learn something from the story page. The highlight is supposed to be an educational glimpse of the full section's content, remaining fully true to the original text.

We provide examples below of both strong and weak summaries for our use case. Note that we expect the summary to contain only factually correct information from what is provided in the original section text. We provide weak summary examples to demonstrate ways the summary **style** can be incorrect.

*Example 1: The Proverb section of the Apple Wikipedia article.*
**SECTION TEXT**
The proverb, "An apple a day keeps the doctor away", addressing the supposed health benefits of the fruit, has been traced to 19th-century Wales, where the original phrase was "Eat an apple on going to bed, and you'll keep the doctor from earning his bread". In the 19th century and early 20th, the phrase evolved to "an apple a day, no doctor to pay" and "an apple a day sends the doctor away"; the phrasing now commonly used was first recorded in 1922. Despite the proverb, a 2015 study found no evidence that eating an apple daily prevents visits to a physician.

STRONG SUMMARIES

✅ The proverb "An apple a day keeps the doctor away" has been traced back to 19th-century Wales, but has yet to be proven scientifically.

✅ "An apple a day keeps the doctor away" originated in Wales in the 19th century with the phrasing "Eat an apple on going to bed, and you'll keep the doctor from earning his bread."

✅ The proverb "An apple a day keeps the doctor away" has had multiple phrasings over the years, first being traced to 19th-century Wales.

WEAK SUMMARIES *These are poor summaries because they use words like "this" or "it", have a dialogue-like style, or overly abstract the factual information from the Wikipedia section.*

✖ This section talks about the phrase "An apple a day keeps the doctor away" and all of the ways it has been said.

✖ It talks about how apples don't actually prevent doctor visits.

✖ The apple proverb has existed for centuries.

*Example 2: The Breeds section of the Dog Wikipedia article.*

**SECTION TEXT**

Dogs are the most variable mammal on earth with around 450 globally recognized dog breeds. In the Victorian era, directed human selection developed the modern dog breeds, which resulted in a vast range of phenotypes. Most breeds were derived from small numbers of founders within the last 200 years, and since then dogs have undergone rapid phenotypic change and were formed into today's modern breeds due to artificial selection imposed by humans. The skull, body, and limb proportions vary significantly between breeds, with dogs displaying more phenotypic diversity than can be found within the entire order of carnivores. These breeds possess distinct traits related to morphology, which include body size, skull shape, tail phenotype, fur type and color. Their behavioral traits include guarding, herding, and hunting, retrieving, and scent detection. Their personality traits include hypersocial behavior, boldness, and aggression, which demonstrates the functional and behavioral diversity of dogs. As a result, present day dogs are the most abundant carnivore species and are dispersed around the world. The most striking example of this dispersal is that of the numerous modern breeds of European lineage during the Victorian era.

STRONG SUMMARIES

✅ Around 450 dog breeds have been globally recognized, making dogs the most variable mammal with significant differences in behavioral traits and physical characteristics.

✅ The breeding of dogs during the Victorian Era resulted in around 450 globally recognized dog breeds from a small number of founders.

✅ Dogs are the most variable mammal on Earth, having significant variance in skull, body, and limb proportions, also differing personality traits like sociability and boldness.

✅ Present day dogs are the most abundant carnivore with around 450 recognized dog breeds.

✅ With around 450 globally recognized breeds and high variance in both physical and behavioral characteristics, dogs display more phenotypic diversity than all other carnivores combined.

 *These are poor summaries because they are either too brief, reduce and abstract the factual information too much, or use words like "this" and "it."*

- ✖ There are many types of dogs on Earth.
- ✖ This section discusses the difference behavioral, physical, and personality traits dog breeds can have.
- ✖ It describes how dog breeding was done to result in 450 breeds and mentions that dogs are the most variable mammal.

When selecting images, we want you to choose the image you see best fit to go with the section content and summary text. It should be topically relevant and the image you feel is most visually appealing.

## UI PLUGIN EXAMPLE (UPDATED)

Read the below section from a Wikipedia page. The first sentence is highlighted in yellow. Does the first sentence provide a strong and concise (fewer than 30 words) summary of the contents of the entire section?

If you need additional context on this section's text and or topic to answer this question, you can click here to see the description of the Wikipedia page. Otherwise, continue with the annotation task.

Section Title: Etymology
Wikipedia Page Title: Apple

The word apple, formerly spelled æppel in Old English, is derived from the Proto-Germanic root *ap(a)laz, which could also mean fruit in general. This is ultimately derived from Proto-Indo-European *ab(e)l-, but the precise original meaning and the relationship between both words[clarification needed] is uncertain.

As late as the 17th century, the word also functioned as a generic term for all fruit other than berries but including nuts—such as the 14th century Middle English word appel of paradis, meaning a banana. This use is analogous to the French language use of pomme.

- ☐ Yes, it well summarizes the whole section

- ☐ No, it does not well summarize the whole section

*IF "HERE" WAS CLICKED ON FOR ADDITIONAL CONTEXT ABOVE, SHOW THE FOLLOWING PAGE DESCRIPTION TEXT:*

An apple is an edible fruit produced by an apple tree (Malus domestica). Apple trees are cultivated worldwide and are the most widely grown species in the genus Malus. The tree originated in Central Asia, where its wild ancestor, Malus sieversii, is still found today. Apples

have been grown for thousands of years in Asia and Europe and were brought to North America by European colonists. Apples have religious and mythological significance in many cultures, including Norse, Greek, and European Christian tradition.

*IF THE ANSWER TO THE ABOVE QUESTION WAS NO, SHOW THE FOLLOWING SUMMARIZATION TASK:*

Please write a single sentence summarizing the Wikipedia section. Keep the summary factually accurate to the original text and summarize the content concisely; try to use 15-30 words at most. The goal is to provide an educational, interesting snippet for a story page of this section.

Section Title: Etymology
Wikipedia Page Title: Apple