# OpenReview forum: "A Suite of Generative Tasks for Multi-Level Multimodal Webpage Understanding"
_EMNLP/2023/Conference — EMNLP 2023 Main_

### Official Review · Reviewer_dKPZ · 2023-07-30

**Soundness:** 5

**Excitement:**

4: Strong: This paper deepens the understanding of some phenomenon or lowers the barriers to an existing research direction.

**Missing References:**

[1] Bertscore: Evaluating text generation with bert, ICLR 2018.

[2] BLEURT: Learning Robust Metrics for Text Generation, ACL 2020.

[3] CLIPScore: A Reference-free Evaluation Metric for Image Captioning, EMNLP 2021.

[4] EMScore: Evaluating Video Captioning via Coarse-Grained and Fine-Grained Embedding Matching, CVPR 2022.

**Paper Topic And Main Contributions:**

This paper introduces a large dataset, WikiWeb2M, for studying multimodal webpage understanding. Additionally, the authors propose a novel attention mechanism, Prefix Global, to enhance performance and efficiency. Extensive experiments and analyses demonstrate the superiority of both the dataset and the proposed method.


**Questions For The Authors:**

(1) Is there any noise present in the dataset parsing process? If so, including those noises and discussing their types will be beneficial.

(2) As mentioned in line 236, the method utilizes the earlier content in a body of text. Have the authors explored other parts, such as the last content in a body of text (which is sometimes the conclusion)?


**Reasons To Accept:**

(1) The paper is very well-written and easy to follow. The organization of the manuscript effectively follows their contributions.

(2) This paper's contributions, including both the dataset and the new attention mechanism, will significantly benefit the development of multimodal webpage understanding.

(3) The experiments and analyses are conducted comprehensively, encompassing ablation studies and efficiency analysis.

(4) Figure 4 demonstrates that the proposed method does not encounter performance degradation as the input sequence length increases, which is a promising solution for handling long texts.


**Reasons To Reject:**

(1) Other evaluation metrics, especially those [1,2] that reflect human preference and those designed for multimodal evaluation [3,4], should be reported (see missing references section).


**Reproducibility:**

3: Could reproduce the results with some difficulty. The settings of parameters are underspecified or subjectively determined; the training/evaluation data are not widely available.

**Reviewer Confidence:**

4: Quite sure. I tried to check the important points carefully. It's unlikely, though conceivable, that I missed something that should affect my ratings.

---

> ### Author Rebuttal · Authors · 2023-08-28
>
> We thank Reviewer dKPZ for the detailed review and clear questions. Thank you for identifying that Prefix Global can “enhance performance and efficiency.” We also agree that “the dataset and new attention mechanism will significantly benefit the development of multimodal webpage understanding.”
>
> We now address your comments and questions:
>
> > [R1] *Additional evaluation metrics.*
>
> We will additionally report BLEURT, CLIPScore, and RefCLIPScore with their references accordingly. As BLEURT uses a trained regression model to compute its scores, it can be time consuming to use for evaluation. Below we report BLEURT and CLIPScore on a random subset of 10k samples for all image captioning experiments in the main paper; additional BLEURT scores for other tasks will be added in an updated version of the draft, too. The results follow the same trends.
>
> **Table 5**
> | Feature Inputs                              | BLEURT            | CLIPScore | RefCLIPScore |
> | ----                                                | ----                    | ----              | ----                   |
> | Text                                              | 0.36                   | 0.6850        | 0.7146            |
> | Text, Title                                     | 0.36                   | 0.6845        | 0.7153             |
> | Text, Title, Struct                          | 0.37                   | 0.6865        | 0.7166             |
> | Text, Title, Caption                       | 0.37                   | 0.6851        | 0.7154             |
> | Text, Title, Struct, Caption            | 0.37                   | 0.6878        | 0.7177             |
> | Text, Title, Struct, Image              | **0.41**              | **0.7340**  | 0.7575             |
> | Tex, Title, Struct, Caption, Image | **0.41**              | 0.7329       | **0.7576**       |
>
> **Table 6**
> | Model | ViT Data| BLEURT | CLIPScore | RefCLIPScore |
> | ----      | ----         | ----                    | ----              | ----                  |
> | Base  | im21k     | **0.41**            | 0.7329        |  0.7576           |
> | Base  | JFT        | 0.4                    | 0.7291        | 0.7534            |
> | Large | im21k     | **0.41**            | **0.7374**  | **0.7611**       |
> | Large | JFT        | **0.41**            | 0.7263        | 0.7527             |
>
> **Table 7**
> | Input Section Type     | Section Source | BLEURT | CLIPScore | RefCLIPScore |
> |              ----                 |           ----          |        ----             |      ------       | ----                   |
> | Target                         | WIT                  | 0.4                    |     0.7287    | 0.7527             |
> | Target, Desc               | WIT                  | 0.4                    |    0.7307     | 0.7537             |
> | Target, Desc, Context | WIT                  | 0.4                   |    0.7325      | 0.7558            |
> | Target, Desc, Context | WikiWeb2M     | **0.41**            | **0.7329**   | **0.7576**       |
>
> > [Q1] *Is there any noise present in the dataset parsing process?*
>
> The text and image content in WikiWeb2M is parsed from the webpage HTML. To the best of our knowledge there should not be any noise present in this part of the data parsing pipeline. For task specific processing, we believe noise may only be introduced in the processing of the first sentences of each section (to be used for either the section summarization task or as global inputs for page description generation).
>
> Specifically, the code used to parse the first sentence may prematurely split a sentence from a period that does not signal the end of the sentence. We did manually inspect samples early on and found this to be rare (*e.g.*, 96/100 random samples were split correctly). Additionally, the sentences which were split prematurely could still be valid standalone sentences. Below we share four examples split prematurely:
>
> | Full First Sentence (By Manual Inspection) | Our Data Processor’s First Sentence |
> | ---------                                                          | -------                                                   |
> | In 2007, Saade was a founding member of What's Up!, a Swedish boy band which included Robin Stjernberg, Ludwig "hejLudde" Keijser and Johan Yngvesson. | In 2007, Saade was a founding member of What's Up! |
> | Unicamp offers over one thousand extension programs to the community, with different levels of minimum requirements (high school degree, undergraduate degree, etc.) and across all areas of study, focusing mainly on specialization courses and community outreach. | Unicamp offers over one thousand extension programs to the community, with different levels of minimum requirements (high school degree, undergraduate degree, etc.) |
> | The proposed biosynthesis of ascofuranone was reported by Kita et al., as well as by Abe et al. | The proposed biosynthesis of ascofuranone was reported by Kita et al. |
> | 1988 The choir sang the German premiere of Joseph Jongen's Mass for choir, brass ensemble and organ, Op. 130, which was not yet in print then, both in the Stiftskirche of Aschaffenburg and in St. Bonifatius. | 1988 The choir sang the German premiere of Joseph Jongen's Mass for choir, brass ensemble and organ, Op. |
>
> We will add pseudo code for our sentence-splitter into the Appendix in case it is of use to anyone as well as a random set of examples from the dataset illustrating the first sentence produced by our code. Additionally, we do include the entire full text of each section in our annotations, which will allow other researchers to reprocess the raw text as they see fit.
>
> >[Q2] *Have the authors explored other parts of the text for prefix tokens, such as the last content in a body of text?*
>
> This is an interesting point and not something we explicitly explored during our experiments. Our choice of utilizing earlier content in a body of text originates from the Lead Bias phenomenon, in which earlier content is typically more important. This phenomenon has historically been used to select more important content [1, 2, 3]. Not all webpages may reliably have a final text section which is conclusion-style, which also motivates using earlier page content.
>
> [1] Kedzie et al. *Content selection in deep learning models of summarization.* EMNLP 2018.
>
> [2] Xing et al. *Demoting the lead bias in news summarization via alternating adversarial learning.* ACL 2021.
>
> [3] Zhu et al. *Leveraging lead bias for zero-shot abstractive news summarization.* SIGIR 2021.

---

### Official Review · Reviewer_pHok · 2023-08-02

**Soundness:** 5

**Excitement:**

4: Strong: This paper deepens the understanding of some phenomenon or lowers the barriers to an existing research direction.

**Paper Topic And Main Contributions:**

The paper introduces the Wikipedia Webpage suite (WikiWeb2M) dataset, which consists of 2 million webpages, to study multimodal webpage understanding. The key feature of WikiWeb2M is that it retains the complete web content, instead of information filtered for a particular use case. WikiWeb2M enables a systematic study of several generative modeling tasks with text as the output: page description generation, section summarization, and contextual image captioning. To address the long context problem, the authors also propose a novel method called Prefix Global, which selects the most relevant image and text content as global tokens to attend to the rest of the webpage for context. The experimental results demonstrate that the new annotations from WikiWeb2M improve task performance compared to prior data. Comprehensive analyses of sequence length, input features, and model size were also conducted.

**Reasons To Accept:**

1. WikiWeb2M is a useful resource for multimodal webpage understanding. The inclusion of both image-caption pairs and long text articles in one place makes it a comprehensive dataset for various generative tasks.
2. The proposed Prefix Global method is well-motivated. It helps lower computational complexity and retain performance.
3. The experimental results demonstrate the utility of WikiWeb2M annotations. Comprehensive analyses were also conducted.

**Reasons To Reject:**

1. It would be interesting to see the performance of LLMs with vision input (e.g., llava) on the proposed dataset.

**Reproducibility:**

4: Could mostly reproduce the results, but there may be some variation because of sample variance or minor variations in their interpretation of the protocol or method.

**Reviewer Confidence:**

4: Quite sure. I tried to check the important points carefully. It's unlikely, though conceivable, that I missed something that should affect my ratings.

---

> ### Author Rebuttal · Authors · 2023-08-28
>
> We thank Reviewer pHok for the thorough review of our paper and for identifying the benefits of both our data and new attention Prefix Global. We agree that a “key feature of WikiWeb2M is that it retains the complete web content, instead of information filtered for a particular use case” and that Prefix Global can help “address the long context problem” present with webpage data.
>
> We address your concern below:
> >[R1] *It would be interesting to see the performance of LLMs with vision input (e.g., LLaVA) on the proposed dataset.*
>
> Our experiments utilize T5 [1] and ViT [2], with three variants of T5 defined by different encoder attention mechanisms (full attention [1], Transient Global [3], and our Prefix Global). T5 was in part chosen to compare our Prefix Global attention to prior work Transient Global [3], as they also used T5. We agree that evaluating our suite of webpage tasks with LLMs pretrained with visual input such as LLaVA would be interesting and leave this to future work.
>
> [1] Raffel et al. *Exploring the limits of transfer learning with a unified text-to-text transformer.* JMLR 2020.
>
> [2] Dosovitskiy et al. *An image is worth 16x16 words: Transformers for image recognition at scale.* ICLR 2021.
>
> [3] Guo et al. *LongT5: Efficient text-to-text transformer for long sequences.* NAACL 2022.

---

### Official Review · Reviewer_X6bk · 2023-08-04

**Soundness:** 4

**Excitement:**

4: Strong: This paper deepens the understanding of some phenomenon or lowers the barriers to an existing research direction.

**Missing References:**

Pauwels, Luc. "A multimodal framework for analyzing websites as cultural expressions." Journal of Computer-Mediated Communication 17.3 (2012): 247-265.

**Paper Topic And Main Contributions:**

The submission introduces three new generative tasks: page description generation, section summarization, and contextual image caption to verify multimodel webpage understanding. A dataset of 2M pages named WikiWeb2M also be introduced. The authors also design a novel attention mechanism Prefix GLobal to select the most relevant image and text content.

**Questions For The Authors:**

What are the contributions of structural data work in different tasks?


As the first sentence can be a reasonable summary, how do you know if the model will directly borrow the first sentence to make the task trivial?

**Reasons To Accept:**

It is well written with detailed analysis.
It provides a large dedicated dataset  of MultimodalWebpage with a trained model on this dataset.

**Reasons To Reject:**

The tasks they proposed are either multi-modal summarization or captioning and don't have much novelty.

The contribution of structural metadata signaling is not clear.

**Reproducibility:**

3: Could reproduce the results with some difficulty. The settings of parameters are underspecified or subjectively determined; the training/evaluation data are not widely available.

**Reviewer Confidence:**

4: Quite sure. I tried to check the important points carefully. It's unlikely, though conceivable, that I missed something that should affect my ratings.

---

> ### Author Rebuttal · Authors · 2023-08-28
>
> We’d like to thank Reviewer X6bk for their time spent reviewing our paper and thoughtful questions. We appreciate that they believed the paper is “well written with detailed analysis” and acknowledge WikiWeb2M to be a “large dedicated dataset of multimodal webpages.”
>
> >[Missing References]
>
> Thank you for identifying this paper. *A multimodal framework for analyzing websites as cultural expressions* provides relevant context for why studying webpages from a multimodal perspective is important; we will update our draft to include the missing reference in related work.
>
> To address your questions:
>
> >[Q1] *What are the contributions of the structural metadata made available by WikiWeb2M?*
>
> The structural metadata made newly available with WikiWeb2M informs the section structure for each webpage; this tells us where each chunk of text and each image is located. This structural information is what allows us to define the Prefix Global attention over our webpage data. *I.e.*, without it, we would not be able to identify which images and section contents are the most relevant for each task. By knowing the section a body of text and set of images come from, we can utilize them as global tokens in our new attention mechanism.
>
> Secondly, we do find that simply including special structure tokens interleaved in the input sequence improves performance for page description generation and contextual image captioning (see Table 5 results, row 4 versus row 5). In future work, we agree that we could use the structural metadata in other ways. For example, it could be interesting to define a local attention whose attention neighborhood is adaptively defined by the tokens which belong to the same section or some other structural metadata available.
>
> >[Q2] *How do we ensure that the model does not borrow the first sentence, making the section summarization task trivial?*
>
> For a given task, we **remove** the text that serves as the target summary or caption from our inputs to the model; this ensures there is no model “cheating.” *E.g.*, for section summarization, since we utilize the first sentence of a section as its pseudo target summary, we remove the first sentence from our inputs to the model and note this on L194 of the paper. We will revise the writing to make this more clear and explicit.

---

### Meta-Review · Area_Chair_7YHj · 2023-09-15

**Recommendation:** 5

**Metareview:**

The paper illustrates a new dataset, WikiWeb2M, designed to study multimodal webpage understanding. This dataset comprises 2 million webpages and retains the entirety of web content, rather than information tailored for specific use cases. The dataset facilitates the exploration of several generative modeling tasks that produce text outputs, including page description generation, section summarization, and contextual image captioning. To address challenges arising from long context, a novel attention mechanism named Prefix Global is proposed. This mechanism selects the most pertinent image and text content as global tokens, which then attend to the rest of the webpage for context. Experimental results indicate that annotations from WikiWeb2M enhance task performance compared to previous data. The studies also encompass comprehensive analyses concerning sequence length, input features, and model size, underscoring the dataset's significance and the efficacy of the Prefix Global method.

The paper's well-structured presentation, introduction of a comprehensive dataset, innovative attention mechanism, and robust experimental results make it a worthy contribution to the field. In particular: a) The paper introduces WikiWeb2M, a dedicated dataset for multimodal webpage understanding. This dataset is unique in its comprehensiveness, combining both image-caption pairs and extensive text articles, making it a valuable resource for a variety of generative tasks. b) The experiments conducted using the WikiWeb2M annotations show promising results. The paper also includes thorough analyses, such as ablation studies and efficiency analysis. Notably, the proposed method exhibits consistent performance, even as input sequence length increases, indicating its potential in handling long texts.

---

### Decision · Program_Chairs · 2023-10-07

**Decision:**

Accept-Main

**Comment:**

The paper illustrates a new dataset, WikiWeb2M, designed to study multimodal webpage understanding. This dataset comprises 2 million webpages and retains the entirety of web content, rather than information tailored for specific use cases. The dataset facilitates the exploration of several generative modeling tasks that produce text outputs, including page description generation, section summarization, and contextual image captioning. To address challenges arising from long context, a novel attention mechanism named Prefix Global is proposed. This mechanism selects the most pertinent image and text content as global tokens, which then attend to the rest of the webpage for context. Experimental results indicate that annotations from WikiWeb2M enhance task performance compared to previous data. The studies also encompass comprehensive analyses concerning sequence length, input features, and model size, underscoring the dataset's significance and the efficacy of the Prefix Global method.

The paper's well-structured presentation, introduction of a comprehensive dataset, innovative attention mechanism, and robust experimental results make it a worthy contribution to the field. In particular: a) The paper introduces WikiWeb2M, a dedicated dataset for multimodal webpage understanding. This dataset is unique in its comprehensiveness, combining both image-caption pairs and extensive text articles, making it a valuable resource for a variety of generative tasks. b) The experiments conducted using the WikiWeb2M annotations show promising results. The paper also includes thorough analyses, such as ablation studies and efficiency analysis. Notably, the proposed method exhibits consistent performance, even as input sequence length increases, indicating its potential in handling long texts.